# MoST: Mixing Speech and Text with Modality-Aware Mixture of Experts

## Abstract

We present MoST (Mixture of Speech and Text), a novel multimodal large language model that seamlessly integrates speech and text processing through our proposed Modality-Aware Mixture of Experts (MAMoE) architecture. While current multimodal models typically process diverse modality representations with identical parameters—disregarding their inherent representational differences, we introduce specialized routing pathways that direct tokens to modality-appropriate experts based on input type. MAMoE simultaneously enhances modality-specific learning and cross-modal understanding through two complementary components: modality-specific expert groups that capture domain-specific patterns and shared experts that facilitate information transfer between modalities. Building on this architecture, we develop an efficient transformation pipeline that adapts the pretrained MoE language model through strategic post-training on ASR and TTS datasets, followed by fine-tuning with a carefully curated speech-text instruction dataset. A key feature of this pipeline is that it relies exclusively on fully accessible, open-source datasets to achieve strong performance and data efficiency. Comprehensive evaluations across ASR, TTS, audio language modeling, and spoken question answering benchmarks show that MoST consistently outperforms existing models of comparable parameter counts. Our ablation studies confirm that the modality-specific routing mechanism and shared experts design significantly contribute to performance gains across all tested domains. To our knowledge, MoST represents the first fully open-source speech-text LLM built on a Mixture of Experts architecture. [1]

## 1 Introduction

The seamless integration of speech and text modalities represents a cornerstone objective in the pursuit of versatile and natural human-computer interaction. Foundation models, particularly Large Language Models (LLMs), have demonstrated remarkable capabilities in text understanding and generation (OpenAI et al., 2024; DeepSeek-AI et al., 2024), motivating efforts to extend their prowess to encompass spoken language. However, unifying inherently different modalities like continuous, high-dimensional speech waveforms and discrete, symbolic text sequences within a single model presents significant architectural and training challenges.

A prevalent approach in multimodal learning involves mapping different modalities into a shared representational space, often processed by a common set of transformer parameters (Nguyen et al., 2024; Défossez et al., 2024). While conceptually simple, forcing diverse modality representations through identical parameters disregards their inherent representational differences. Specifically for speech and text, this uniform treatment risks representational interference, where the distinct statistical properties and structural nuances of each modality are diluted. Furthermore, it can hinder the development of highly specialized processing pathways necessary to capture the fine-grained details unique to each domain—such as phonetic variations in speech or complex semantic relations in text.

Mixture of Experts (MoE) architectures have emerged as a powerful technique for scaling model capacity while maintaining computational efficiency (Shazeer et al., 2017; Lepikhin et al., 2020;

---

[1] We release MoST model, training code, inference code, and training data at https://github.com/anonymous29008/MoST/tree/main

Fedus et al., 2022). By sparsely activating only a subset of parameters (experts) for each input token, MoE models can significantly increase their parameter count without a proportional increase in inference cost. While promising for handling complex data, standard MoE routing mechanisms are typically modality-agnostic; tokens are routed based on learned gating functions over their representations, without explicit consideration of their originating modality. This leaves untapped the potential to dedicate specialized expert capacity based on known input characteristics.

In this work, we propose MoST (Mixture of Speech and Text), a novel speech-text LLM built upon a Modality-Aware Mixture of Experts (MAMoE) architecture. MAMoE directly addresses the limitations of uniform parameter processing by incorporating two principal components: **modality-specific expert groups** (e.g., text-only, audio-only) governed by a modality-aware routing mechanism, and **shared experts** accessible to all modalities, which facilitate cross-modal interaction. The modality-aware routing explicitly leverages token modality information to direct inputs towards appropriate modality-specific experts. Text tokens are primarily routed to text and shared experts, while audio tokens target audio and shared experts. This design allows for: (1) Specialized Learning: Modality-specific experts can capture the intricate patterns unique to either speech or text without interference. (2) Efficient Cross-Modal Interaction: Shared experts act as a bridge, facilitating knowledge transfer and cross-modal alignment crucial for tasks like Automatic Speech Recognition (ASR) and Text-to-Speech (TTS) synthesis.

To effectively realize MoST, we develop an efficient LLM to Speech-Text Transformation Pipeline. Starting from a pretrained MoE LLM, we first conduct targeted post-training on large ASR and TTS datasets. This stage primes the modality-specific experts and adapts the shared experts for cross-modal tasks. Subsequently, we perform instruction fine-tuning using a carefully curated dataset comprising diverse speech and text instructions, enhancing the model's versatility and controllability. A key aspect of our approach is the exclusive use of fully accessible, open-source training data to achieve strong performance. This strategic choice distinguishes MoST from some prominent speech-text models, such as SpiritLM (Nguyen et al., 2024) and Moshi (Défossez et al., 2024), whose development incorporated datasets that are not publicly accessible.

Comprehensive evaluations across a range of speech-text benchmarks, including Automatic Speech Recognition (ASR), Text-to-Speech (TTS) synthesis, audio language modeling, and spoken question answering, demonstrate that MoST achieves state-of-the-art or competitive performance compared to existing models with similar parameter counts. Our ablation studies rigorously validate the effectiveness of the MAMoE architecture, confirming that the modality-aware routing mechanism is a key contributor to performance gains across both modality-specific and cross-modal tasks. To the best of our knowledge, MoST is the first fully open-source speech-text LLM leveraging a Mixture of Experts architecture with modality-specific routing.

Our main contributions are:

- **Modality-Aware Mixture of Experts (MAMoE)**: A novel multimodal MoE architecture that directs tokens to modality-specific expert groups and shared experts, enhancing specialized learning and cross-modal interaction.
- **Efficient Text-Speech Transformation Pipeline**: A data-efficient pipeline for adapting a pretrained LLM into a speech-text LLM via targeted post-training and instruction tuning.
- **Open Release of MoST Model**: We release the MoST model weights, training and inference code, and training dataset to facilitate future research and development in multimodal AI.

## 2 RELATED WORK

**End-to-end Speech-Text Modeling**. Existing end-to-end speech-text models like Qwen2-Audio (Chu et al., 2024) (text output only) and AudioLM (Borsos et al., 2023) (no direct text integration) have specific limitations. Others such as SpiritLM (Nguyen et al., 2024) and Moshi (Défossez et al., 2024) employ dense transformers, which process all tokens uniformly, unlike MoST's Modality-Aware Mixture of Experts (MAMoE), which uses specialized and shared experts for modality-specific processing and cross-modal understanding.

**Mixture of Experts**. Standard Mixture of Experts (MoE) architectures (Lepikhin et al., 2020; Fedus et al., 2022; Du et al., 2022) scale models efficiently but typically feature modality-agnostic routing,

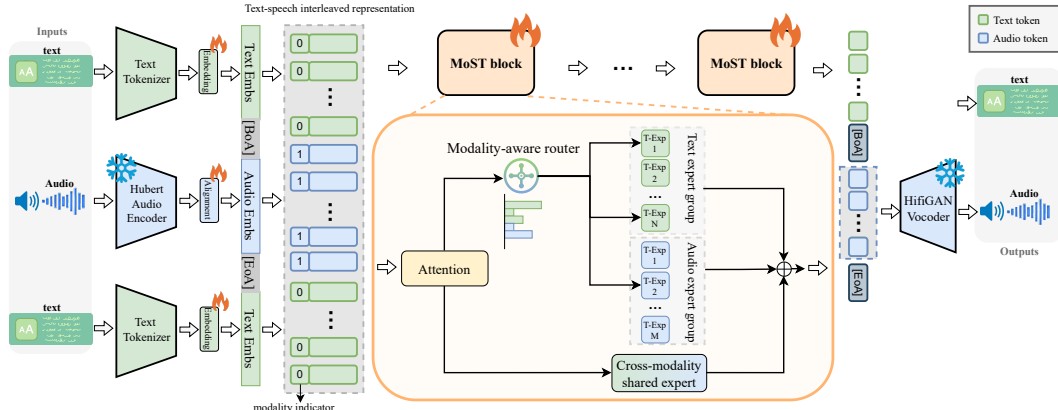

Figure 1: MoST overall architecture. MoST supports interleaved text and audio as input and can generate both text and audio. MoST blocks are transformer blocks featured with Modality-Aware Mixture of Experts (MAMoE). MAMoE is comprised of three critical components: modality-specific expert groups, cross-modal shared expert, and the modality-aware router. Modality-aware router utilizes the modality indicator information for modality-specific routing.

limiting their multimodal effectiveness. While some research adapts MoE for general multimodal tasks (e.g., image-text) by grouping experts (Lin et al., 2024; Shen et al., 2024), MoST is the first to introduce MAMoE specifically for speech-text. Another key difference is that MoST's MAMoE employs shared experts for cross-modal interaction, which is novel.

**Multimodal Adaptation from LLMs**. LLMs are often adapted for multimodality via fine-tuning (Zhang et al., 2024), instruction tuning (Liu et al., 2023), or speech-specific methods (Fathullah et al., 2023). MoST presents a unique transformation pipeline: starting from a pretrained MoE LLM, it undergoes targeted ASR/TTS post-training, followed by fine-tuning with a curated speech-text instruction dataset. This approach offers significant data efficiency over conventional adaptation strategies.

## 3 MoST: MIXTURE OF SPEECH AND TEXT

### 3.1 MODEL OVERVIEW

MoST (Mixture of Speech and Text) is a unified foundation model built on a Mixture of Experts (MoE) architecture for seamless speech and text processing. Unlike discrete tokenization approaches (Borsos et al., 2023; Nguyen et al., 2024), MoST directly processes continuous audio waveforms via audio encoder $E_{audio}$: $\mathbf{H}_{audio} = E_{audio}(\mathbf{A}) \in \mathbb{R}^{L_{audio} \times d_{model}}$. Text representations are obtained through standard tokenization and embedding: $\mathbf{H}_{text} = E_{text}(\mathbf{T}) \in \mathbb{R}^{L_{text} \times d_{model}}$. These modalities are combined into unified input $\mathbf{H}_{input}$.

The core transformer decoder $D_{MoST}$, adapted from a pretrained MoE LLM, incorporates our Modality-Aware Mixture of Experts (MAMoE) in designated layers. MAMoE routes tokens to modality-specific experts and shared experts for cross-modal interaction: $\mathbf{H}_{output} = D_{MoST}(\mathbf{H}_{input})$.

MoST generates both text via language model head $P_{text}$ and audio via vocoder $G_{audio}$ conditioned on $\mathbf{H}_{output}$ and speaker embeddings, enabling diverse end-to-end speech-text tasks. Figure 1 illustrates the architecture.

### 3.2 AUDIO ENCODER AND DECODER

MoST processes continuous audio waveforms directly, preserving richer acoustic information than discrete tokenization methods (Borsos et al., 2023; Nguyen et al., 2024).

**Audio Encoder** Input waveforms $\mathbf{A}$ are encoded using frozen HuBERT (Hsu et al., 2021), yielding features $\mathbf{F}_{hubert} \in \mathbb{R}^{L_{audio} \times d_{hubert}}$, then projected to model dimension:

$$\mathbf{H}_{audio} = \mathbf{F}_{hubert} \mathbf{W}_{proj} \in \mathbb{R}^{L_{audio} \times d_{model}}. \tag{1}$$

These continuous embeddings replace placeholder tokens in the input sequence.

**Audio Decoder** For synthesis, MoST employs HifiGAN vocoder $G_{audio}$ (Kong et al., 2020), generating waveforms $\hat{\mathbf{A}}$ conditioned on predicted HuBERT tokens $\mathbf{C}_{audio} = P_{hubert\_tokens}(\mathbf{H}_{output})$ and speaker embeddings $\mathbf{s}$:

$$\hat{\mathbf{A}} = G_{audio}(\mathbf{C}_{audio}, \mathbf{s}). \tag{2}$$

---

**Algorithm 1** Modality-Aware Expert Routing

---

**Require:** Token representation $\mathbf{h}_t$, modality indicator $\mathbf{m}_t$, gating weights $\mathbf{W}_g$, expert groups $\mathcal{E}_{text}$, $\mathcal{E}_{audio}$, number of experts to select $\mathbf{K}$

**Ensure:** Routed output $\mathbf{y}_{routed,t}$

1: **Step 1: Compute Raw Gating Scores**
2: $\mathbf{s}_t \leftarrow \text{softmax}(\mathbf{h}_t W_g) \in \mathbb{R}^N$ $\qquad\qquad\qquad$ ▷ Raw scores over all $N$ experts
3: **Step 2: Create Modality-Specific Mask**
4: **for** $i = 1$ to $N$ **do**
5: $\quad$ **if** $m_t = 0$ (text) and $E_i \in \mathcal{E}_{text}$ **then**
6: $\qquad$ $M_{m_t,i} \leftarrow 1$
7: $\quad$ **else if** $m_t = 1$ (audio) and $E_i \in \mathcal{E}_{audio}$ **then**
8: $\qquad$ $M_{m_t,i} \leftarrow 1$
9: $\quad$ **else**
10: $\qquad$ $M_{m_t,i} \leftarrow 0$
11: $\quad$ **end if**
12: **end for**
13: **Step 3: Apply Modality Mask**
14: $\mathbf{s}'_t \leftarrow \mathbf{s}_t \odot \mathbf{M}_{m_t}$ $\qquad\qquad\qquad\qquad$ ▷ Element-wise multiplication
15: **Step 4: Select Top-K Experts**
16: $\mathcal{I}_t \leftarrow \text{TopK}(\mathbf{s}'_t, K)$ $\qquad\qquad\qquad\qquad$ ▷ Indices of top-K experts
17: $\mathbf{w}'_t \leftarrow \{\mathbf{s}'_t[i] : i \in \mathcal{I}_t\}$ $\qquad\qquad\qquad$ ▷ Corresponding weights
18: **Step 5: Compute Routed Output**
19: $\mathbf{y}_{routed,t} \leftarrow \sum_{k=1}^{K} w'_{t,k} \cdot E_{i_{t,k}}(\mathbf{h}_t)$ $\qquad\qquad$ ▷ Weighted expert outputs
20: **return** $\mathbf{y}_{routed,t}$

---

### 3.3 Modality-Aware Mixture of Experts (MAMoE)

#### 3.3.1 Modality-Specific Expert Groups

In the MAMoE layers, the set of $N$ available routed experts, denoted $\mathcal{E} = \{E_1, E_2, \ldots, E_N\}$, is partitioned into disjoint modality-specific groups. Based on the primary modalities in MoST (speech and text), we define: **Text Expert Group** ($\mathcal{E}_{text}$), a subset of experts designated to primarily process text-derived token representations, and **Audio Expert Group** ($\mathcal{E}_{audio}$), a subset of experts designated to primarily process audio-derived token representations (specifically, features derived from the HuBERT encoder).

These sets are disjoint, $\mathcal{E}_{text} \cap \mathcal{E}_{audio} = \emptyset$, and their union constitutes the full set of routed experts, $\mathcal{E}_{text} \cup \mathcal{E}_{audio} = \mathcal{E}$. Each expert $E_i$ is an MLP, consisting of gate, up, and down projection layers. This partitioning allows experts to specialize in capturing the distinct statistical patterns and representations characteristic of either speech or text.

#### 3.3.2 Cross-Modal Shared Experts

While modality-specific experts promote specialized processing, effective multimodal understanding requires mechanisms for information integration across modalities. In MoST, this is facilitated by incorporating shared experts, implemented as a separate, parallel MLP block ($E_{shared}$). Unlike the routed experts $\mathcal{E}$, the shared expert block processes *all* tokens passing through the MAMoE layer.

Given the input hidden state $\mathbf{h}$ to the MAMoE layer, the output $\mathbf{y}_{routed}$ from the modality-specific routed experts (detailed in Sec 3.3.3) is combined with the output of the shared expert block:

$$\mathbf{y}_{mamoe} = \mathbf{y}_{routed} + E_{shared}(\mathbf{h}). \tag{3}$$

This parallel shared expert acts as a common pathway, allowing for the exchange and integration of information learned from different modalities, complementing the specialized processing occurring within the modality-specific expert groups.

### 3.3.3 MODALITY-AWARE ROUTER

MAMoE's core routing mechanism directs each token $\mathbf{h}_t$ to modality-specific experts $E_i \in \mathcal{E}$ based on its content and modality indicator $m_t$ (e.g., $m_t = 0$ for text, $m_t = 1$ for audio). Algorithm 1 presents the step-by-step procedure for computing modality-aware routing scores.

During training, an auxiliary load balancing loss, based on $\mathbf{s}'_t$, promotes balanced expert utilization within each modality group. This modality-aware routing directs tokens to specialized pathways, enhancing expert capacity use and modality-specific representation learning.

## 4 EFFICIENT TRAINING RECIPE OF MoST

We develop an efficient training recipe to transform a MoE LLM into MoST. Our approach consists of carefully designed data preparation procedures followed by a two-stage training protocol. Comprehensive details of all procedures are provided in Appendix A.

### 4.1 DATA PREPARATION

**ASR and TTS Data Curation.** We curate large-scale ASR and TTS datasets from open-source resources, primarily utilizing Common Voice (Ardila et al., 2020b) and LibriHeavy (Kang et al., 2024). For ASR data, we use the standard format with speech input and text transcription output. For TTS data, we reverse the modalities to create paired speech-text samples.

**Speech-Text Instruction Dataset Construction.** We construct a speech-text instruction-following dataset through two complementary strategies: (1) **Interrupted Dialogue Synthesis**: We augment existing dialogue datasets (Allal et al., 2025) with realistic interruptions using LLM-based synthesis to capture natural conversational dynamics and turn-taking patterns. (2) **Text-to-Speech Instruction Conversion**: We convert text-based instruction datasets into speech formats using our intermediate TTS-trained MoST, significantly expanding speech instruction data without manual collection.

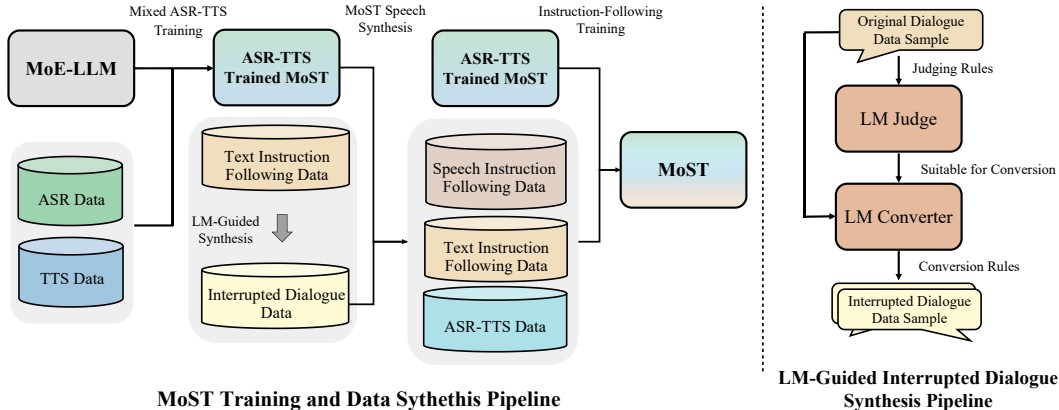

**MoST Training and Data Sythethis Pipeline**

**LM-Guided Interrupted Dialogue Synthesis Pipeline**

Figure 2: MoST training recipe overview. The left panel illustrates the systematic transformation pipeline from MoE LLM to MoST, encompassing data preparation and two-stage training protocol. The right panel details the interrupted dialogue synthesis process for instruction dataset construction.

## 4.2 TRAINING PROTOCOL

**Stage 1: Cross-Modal Post-Training.** We initialize MoST from DeepSeek-v2 Lite (DeepSeek-AI et al., 2024), a pretrained MoE LLM. The initialized model undergoes targeted post-training on the curated ASR and TTS datasets. This stage serves dual purposes: priming modality-specific experts within MAMoE for speech processing and adapting shared experts for fundamental cross-modal alignment between speech and text representations.

**Stage 2: Mixed Instruction Fine-Tuning.** The model is fine-tuned using our constructed speech-text instruction dataset, strategically mixed with a substantial portion of ASR/TTS data from Stage 1. This mixed-task learning approach enhances complex instruction-following capabilities while preventing catastrophic forgetting of foundational speech processing abilities through continual exposure to core tasks.

## 5 EMPIRICAL RESULTS

### 5.1 EVALUATION SETTINGS

To rigorously assess the performance of MoST, we establish a comprehensive evaluation framework encompassing standard benchmarks and strong baseline models.

**Baselines.** We compare MoST against several state-of-the-art open-source speech and text models, including **AudioLM** (Borsos et al., 2023), a prominent model for high-fidelity audio generation; **SpeechGPT** (Zhang et al., 2023), an LLM adapted for speech tasks; **SpiritLM** (Nguyen et al., 2024), a recent speech-language model with strong performance; **Moshi** (Défossez et al., 2024), another relevant model in speech processing; **Qwen2-Audio** (Chu et al., 2024), a powerful open-source model which has shown strong performance in various speech tasks; **Phi-4 Multimodal** (Microsoft et al., 2025), a compact yet powerful multimodal language model that achieves very strong ASR and AST performance; **SeamlessM4T-v2** (Communication et al., 2023), an advanced model for speech translation and other speech-related tasks; **MinMo** (Chen et al., 2025), a recent multimodal model for speech and text; and **LLaMA-Omni2** (Fang et al., 2025), a recent omnimodal language model with speech capabilities.

**Benchmarks and Metrics.** Evaluation spans ASR, TTS, audio language modeling (ALM), and spoken question answering (SQA). For ASR/TTS, we use LibriSpeech-clean and -other (Kang et al., 2024), VoxPopuli-V1.0-en (Wang et al., 2021), and Common Voice 15-en (WER for ASR, TTS; lower is better) (Ardila et al., 2020a). ALM is assessed on sWUGGY, sBLIMP (Both from Nguyen et al. (2020)); sTopic-StoryCloze and sStoryCloze (Both from Hassid et al. (2024)) (accuracy). SQA uses Llama Q (Nachmani et al., 2024) (S→T, S→S), Trivial QA (Joshi et al., 2017) (S→T, S→S), and WebQ (Berant et al., 2013) (S→T, S→S) (task-specific scores/accuracy).

### 5.2 EVALUATION ON TTS AND ASR TASKS

We evaluate MoST on ASR and TTS across multiple datasets in Table 1. MoST demonstrates competitive ASR performance, achieving 2.0% WER on LS-Clean and 3.7% WER on LS-Other, while excelling in challenging cross-dataset scenarios with 6.2% WER on VoxPopuli-V1.0-en and 8.4% WER on Common Voice 15-en. In TTS synthesis, MoST achieves state-of-the-art performance across multiple datasets: 6.0% WER on LS-Clean, 10.1% CER on VoxPopuli-V1.0-en, and 11.5% CER on Common Voice 15-en, consistently outperforming all baselines including the newly added MinMo and LLaMA-Omni2. These results highlight MAMoE's effectiveness in cross-modal tasks and our pipeline's role in creating a versatile speech-text model with strong generalization capabilities.

### 5.3 EVALUATION ON AUDIO LANGUAGE MODELING TASKS

MoST and baseline models are evaluated on four audio language modeling benchmarks—sWUGGY, sBLIMP(Lakhotia et al., 2021), sTopic-StoryCloze, and sStoryCloze(Hassid et al., 2023) —designed to probe audio language modeling capabilities such as grammatical acceptability and discourse coherence. We report accuracies derived from negative log-likelihood (NLL) scores normalized by sequence length, ensuring a fair comparison across variable-length inputs and tasks.

Table 1: ASR (WER % ↓) and TTS (WER % ↓) performance comparison on LibriSpeech clean and other test sets, VoxPopuli-V1.0-en, and Common Voice 15-en. Lower values are better. **MoST (Ours)** results are highlighted with green background.

| Model | LS-Clean | | LS-Other | | VoxPopuli-V1.0-en | | Common Voice 15-en | |
|---|---|---|---|---|---|---|---|---|
| | ASR | TTS | ASR | TTS | ASR | TTS | ASR | TTS |
| SpeechGPT | 11.0 | 14.1 | 16.7 | 15.3 | 18.2 | 21.3 | 19.4 | 23.2 |
| AudioLM | 9.5 | 9.2 | 12.0 | 12.1 | 15.0 | 22.1 | 17.6 | 25.1 |
| SpiritLM | 6.0 | 6.7 | 11.0 | 9.5 | 14.3 | 19.4 | 15.4 | 22.4 |
| Moshi | 5.5 | 7.0 | 12.0 | 7.2 | 8.8 | 10.6 | 9.4 | 14.2 |
| Qwen2-Audio | **1.8** | / | 3.6 | / | 7.1 | / | 8.6 | / |
| Phi-4 Multimodal | 2.1 | 4.8 | **3.5** | 4.1 | 6.3 | 11.5 | 9.2 | 10.8 |
| SeamlessM4T-v2 | 3.3 | 6.3 | 4.2 | **5.3** | 7.5 | 10.3 | 10.3 | 12.1 |
| MinMo | **1.8** | 6.7 | 3.9 | 7.5 | 6.7 | 10.9 | 8.0 | 13.5 |
| LLaMA-Omni2 | 3.5 | 10.1 | 4.0 | 9.2 | 9.5 | 12.4 | 11.3 | 17.2 |
| **MoST (Ours)** | 2.0 | **6.0** | 3.7 | 7.2 | **6.2** | **10.1** | 8.4 | **11.5** |

Table 2: Performance on Audio Language Modeling Tasks. MoST demonstrates superior performance across most benchmarks compared to existing models. **MoST (Ours)** results are highlighted with green background and improvement notes.

| Model | sWUGGY | sBLIMP | sTopic-StoryCloze | sStoryCloze | Average |
|---|---|---|---|---|---|
| AudioLM | 71.50 | **64.70** | - | - | - |
| SpeechGPT | 51.82 | 49.75 | 60.13 | 53.13 | 53.71 |
| spiritLM | 40.14 | 48.28 | 83.32 | 58.95 | 57.67 |
| Moshi | 51.14 | 53.31 | 46.34 | 45.16 | 48.99 |
| Phi-4 Multimodal | 71.84 | 60.21 | 81.55 | 62.39 | 69.00 |
| MinMo | 68.59 | 55.43 | 75.43 | 61.29 | 65.19 |
| LLaMA-Omni2 | 73.21 | 53.59 | 78.21 | 68.55 | 68.39 |
| **MoST (Ours)** | **75.28**↑2.1% | 63.42 | **83.64**↑0.3% | 65.43 | **71.94**↑2.9% |

As shown in Table 2, MoST achieves competitive performance with an average score of 71.18, comparable to Phi-4 Multimodal (71.84) and outperforming MinMo (65.19) and LLaMA-Omni2 (68.39). MoST achieves state-of-the-art performance on sTopic-StoryCloze (83.64), demonstrating strong performance across multiple benchmarks. This strong performance underscores MAMoE's efficacy in learning nuanced audio-specific linguistic patterns for enhanced spoken language understanding.

## 5.4 Evaluation on Spoken Question Answering Tasks

MoST's Spoken Question Answering (SQA) capabilities are evaluated on Llama Q (Nachmani et al., 2024) (S→T and S→S), Trivial QA (Joshi et al., 2017) (S→T and S→S), and WebQ (Berant et al., 2013) (S→T and S→S). As shown in Figure 3, MoST demonstrates strong performance across all SQA tasks, achieving the best or competitive results in most settings. On Llama Q, MoST achieves 74.8 (S→T) and 62.6 (S→S), demonstrating robust speech-to-text understanding while maintaining competitive speech-to-speech performance. On Trivial QA, MoST scores 43.5 (S→T) and a notably high 32.1 (S→S), substantially exceeding most competitors. On WebQ, MoST achieves the best performance with 58.2 (S→T) and 44.7 (S→S), significantly outperforming all baselines including MinMo and LLaMA-Omni2. This strong performance, particularly in challenging S→S settings, highlights MoST's robust end-to-end speech understanding and generation, attributed to the MAMoE architecture.

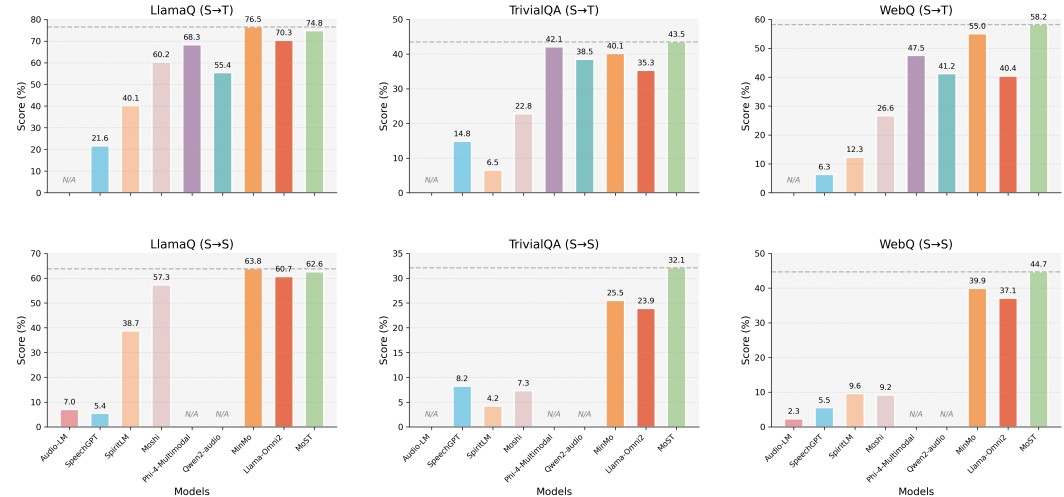

Figure 3: Performance comparison on Spoken Question Answering tasks. MoST achieves competitive or superior performances on all 3 datasets.

## 6 ABLATION STUDY: ARCHITECTURE BENEFITS WITH CONTROLLED COMPARISONS

### 6.1 CONTROLLED INITIALIZATION COMPARISON

To rigorously isolate the architectural contributions of MAMoE from LLM initialization quality, we conduct a controlled comparison using identical Llama3.2 3B (Grattafiori et al., 2024) initialization across three architectural variants: **Dense Baseline** (standard transformer without MoE), **Traditional MoE Upcycling** (Komatsuzaki et al., 2023) (modality-agnostic routing where all tokens compete for all experts), and **MoST-Style Upcycling** (the proposed MAMoE with modality-specific expert groups and shared experts). This controlled experimental design ensures that performance differences stem solely from architectural choices rather than varying initialization quality.

Figure 4 demonstrates the clear architectural progression benefits. Both MoE variants significantly outperform the dense baseline when starting from identical initialization, confirming MoE's fundamental effectiveness for multimodal tasks. Most importantly, our MoST-style upcycling substantially surpasses traditional upcycling across all evaluation metrics, providing compelling evidence for MAMoE's architectural advantages independent of initialization quality. The consistent performance gains—ranging from 7.3% improvement in ASR to 21.8% in SQA—validate that our proposed modality-aware routing mechanism delivers robust benefits regardless of the starting model. While stronger initialization (DeepSeek-V2 Lite) yields superior absolute performance for the full MoST model, this controlled comparison establishes that our architectural innovations provide consistent improvements across diverse speech-text tasks.

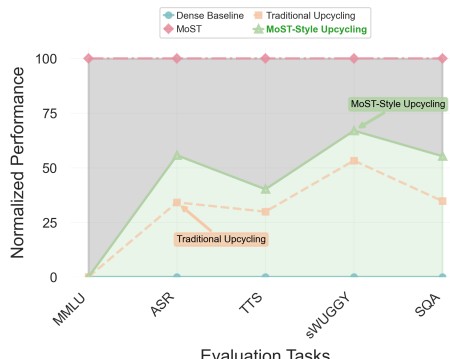

Figure 4: Controlled comparison with identical Llama3.2 3B initialization. We report (100%-WER) for ASR and TTS tasks and accuracy for other tasks.

### 6.2 MAMoE DESIGN VARIANTS ANALYSIS

To validate the individual contributions of MAMoE's core components—modality-aware routing and shared experts—we conduct systematic ablations. We compare MoST against two variants: **MoST-**

**VanillaMoE** (standard MoE routing where all tokens compete for all experts) and **MoST-NoShared** (MAMoE routing but without shared experts, making all experts modality-specific). Both variants maintain the same total number of routed experts as full MoST for fair comparison, precisely isolating the contributions of modality-specific routing and shared expert capacity.

All three models were trained on the combined speech-text instruction following dataset for 10,000 steps (batch size 256). As illustrated in Figure 5, the full MoST model consistently outperforms both ablated variants across the comprehensive task suite. The comparison with MoST-VanillaMoE highlights the significant benefits of MAMoE's modality-specific routing mechanism, while MoST's advantage over MoST-NoShared underscores the crucial role of shared experts in enhancing model generalization and facilitating cross-modal understanding. The training and validation loss curves (Figure 5, top row) further corroborate these findings, showing more favorable learning dynamics for the full MAMoE architecture.

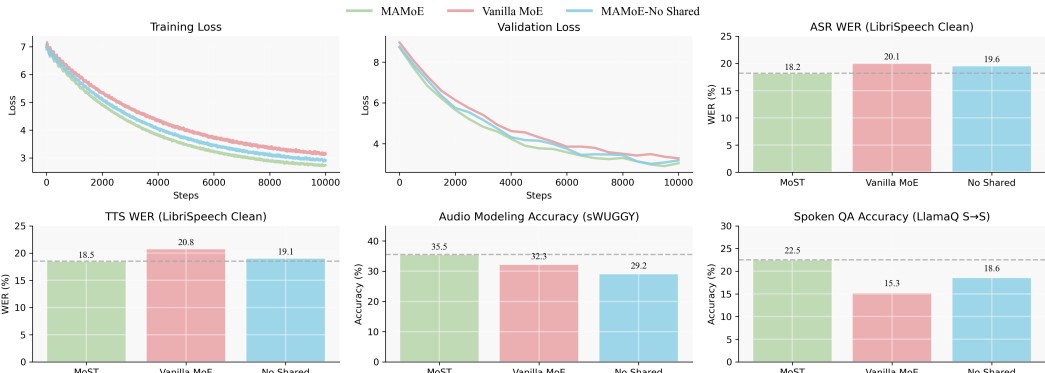

Figure 5: MAMoE design variants analysis. The 2×3 grid displays: Training loss curves; Validation loss curves; ASR, TTS performance (WER) on LibriSpeech dev-clean; Audio language modeling performance (Accuracy) on sWUGGY; Spoken QA performance (Accuracy) on LlamaQ. All metrics are reported at 10,000 training steps. Full MAMoE consistently outperforms ablated versions.

### 6.3 EXPERT ROUTING ANALYSIS

To provide deeper insights into MAMoE's routing mechanism, we analyze expert utilization patterns and quantitative routing metrics. Figure 6 presents a comprehensive visualization combining routing frequency heatmaps (left) with quantitative routing metrics (right) for MoST with MAMoE versus vanilla MoE.

The routing heatmaps reveal that MAMoE exhibits distinct modality-specific routing patterns that become increasingly pronounced during training. Text and audio tokens are systematically directed to their respective expert groups, fostering specialized processing pathways. The quantitative analysis (right panel) confirms that MAMoE achieves progressively lower routing entropy during training, indicating enhanced expert specialization. Besides, MAMoE maintains superior load balancing with consistently lower Gini coefficients compared to vanilla MoE, demonstrating the effectiveness of our routing design.

## 7 CONCLUSION

We introduced MoST, a novel multimodal LLM integrating speech and text with a Modality-Aware Mixture of Experts architecture. MAMoE's modality-specific routing and shared experts enhance specialized processing and cross-modal interaction. Our efficient pipeline transforms a pretrained MoE LLM into a data-efficient speech-text model. Empirical evaluations confirm MoST achieves state-of-the-art or competitive results across ASR, TTS, audio language modeling, and SQA, outperforming strong baselines.

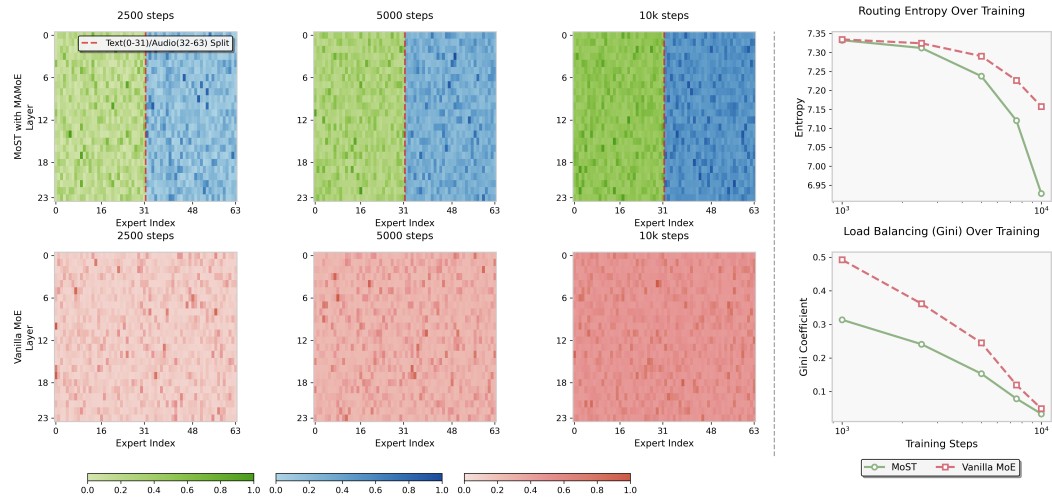

Figure 6: Expert routing analysis for MAMoE versus vanilla MoE. **Left:** Routing frequency heatmaps showing MAMoE's modality-specific routing (top) versus vanilla MoE's diffuse patterns (bottom) at 2,500, 5,000, and 10,000 training steps. **Right:** Quantitative metrics showing MAMoE's superior specialization (lower entropy) and load balancing (lower Gini coefficient) throughout training.

While our index-based 50% expert partition proves effective, it represents a simple initialization strategy. More sophisticated approaches—such as clustering-based or knowledge-preserving partitioning—offer promising directions for future modality-aware MoE research (detailed in Appendix D).

We release MoST's model checkpoints, codebase, and curated data to encourage further research and development in efficient and effective multimodal models.

## REPRODUCIBILITY STATEMENT

To ensure transparency and facilitate reproducibility, we release the full resources required to replicate our experiments. Specifically, we provide the MoST model implementation, including both training and inference code, as well as the complete training datasets, at https://github.com/anonymous29008/MoST/tree/main.

All model configurations used for MoST are reported in Appendix C, detailing architecture choices, hyperparameters, and training schedules. The released code includes scripts for data preprocessing, model training, and evaluation, enabling end-to-end reproduction of our results without additional modifications.

We have verified that running the released code with the provided configurations reproduces the main results presented in the paper. Furthermore, instructions are included in the repository to guide users through environment setup, dependency installation, and hardware requirements.

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

# A DETAILED TRAINING RECIPE

To effectively realize MoST, we develop an efficient LLM-to-Speech-Text Transformation Pipeline. Starting from a pretrained MoE LLM, we first conduct targeted post-training on large ASR and TTS datasets. This stage primes the modality-specific experts and adapts the shared experts for cross-modal tasks. Subsequently, we perform instruction fine-tuning using a carefully curated speech-text instruction dataset. Our training strategy involves mixed-task learning and incorporates techniques to prevent catastrophic forgetting of foundational abilities during fine-tuning. Critically, this pipeline demonstrates remarkable data efficiency, achieving strong performance with significantly fewer training tokens compared to prominent open-source speech-text models like SpiritLM (Nguyen et al., 2024) and Moshi (Défossez et al., 2024).

Table 3 provides a comprehensive overview of our two-stage training protocol, including key hyper-parameters, task mixing ratios, and dataset composition details for each stage.

Table 3: Training Configuration Overview. Detailed breakdown of training stages, configurations, task mixing ratios, and dataset compositions.

| Training Stage | Configuration | Task Mixing Ratios | Dataset Composition Details |
|---|---|---|---|
| **Stage 1:** Cross-Modal Post-Training | **Steps:** 500k **Batch Size:** 512 | **ASR:** 0.4 **TTS:** 0.4 **Text LM:** 0.2 | **ASR/TTS:** LibriHeavy (0.6), Common Voice (0.2), VoxPopuli (0.2) **LM:** RefinedWeb (to prevent forgetting) |
| **Stage 2:** Mixed Instruction Tuning | **Steps:** 10k **Batch Size:** 128 | **Speech Instruct:** 0.4 **Text Instruct:** 0.4 **ASR:** 0.1 **TTS:** 0.1 | **Speech Instruct:** Synthesized from SmolTalk (via MoST Stage 1) **Text Instruct:** Standard open-source instruction data |

## A.1 ASR AND TTS DATA PREPARATION (STAGE 1)

The initial stage of our LLM-to-Speech-Text transformation pipeline involves post-training on large-scale Automatic Speech Recognition (ASR) and Text-to-Speech (TTS) datasets. This step is crucial for priming the modality-specific experts within the MAMoE architecture and adapting the shared experts for fundamental cross-modal alignment tasks.

We utilize two primary open data sources for this stage: the Common Voice corpus (Ardila et al., 2020b) and the full LibriHeavy dataset (Kang et al., 2024), selected for their scale and diversity in speech patterns and linguistic content. The ASR data follows the standard format where the input consists of speech (represented as acoustic features or tokens) and the output is the corresponding text transcription, guided by an instruction like "Transcribe the following speech into text."

To generate the necessary TTS data, we systematically transform the ASR datasets. For each ASR sample {instruction: "Transcribe...", input: speech, output: text}, we create a corresponding TTS sample by reversing the input and output modalities and modifying the instruction. The resulting TTS sample format is {instruction: "Convert the following text into speech.", input: text, output: speech}. This conversion process, (further details were originally noted to be in supplementary materials, which is consistent with an appendix location), ensures that the model is trained on paired speech-text data for both recognition and synthesis tasks, effectively leveraging existing ASR resources to create aligned TTS training data. This phase prepares the model for more complex instruction-following tasks in the subsequent fine-tuning stage.

## A.2 SPEECH-TEXT INSTRUCTION-FOLLOWING DATA CONSTRUCTION (STAGE 2)

Following the initial post-training phase, MoST undergoes instruction fine-tuning to enhance its ability to follow complex commands and engage in nuanced interactions involving both speech and text. This requires a diverse dataset of speech-text instructions. We construct this dataset through two

complementary strategies: synthesizing interrupted dialogues and converting text-based instructions into a speech format.

### A.2.1 INTERRUPTED DIALOGUE SYNTHESIS PIPELINE

Standard instruction datasets often lack the dynamic turn-taking and interruptions characteristic of natural human conversation. To address this, we develop a pipeline to augment existing dialogue datasets with realistic interruptions. We utilize the open-source SmolTalk dataset (Allal et al., 2025) as our base, which contains multi-turn user-assistant conversations.

Our synthesis pipeline employs a large language model (LLM) in a two-step process. First, the LLM analyzes each dialogue from the text instruction following dataset to determine its suitability for adding a user interruption. Suitability criteria include clear role separation (user/assistant), the presence of longer assistant turns amenable to interruption, and contextual appropriateness for an interruption. Second, for dialogues identified as suitable, the LLM is prompted to rewrite the conversation by inserting a natural user interruption during one of the assistant's longer responses. This interruption typically takes the form of a brief clarifying question related to the assistant's immediately preceding statement. The LLM is instructed to modify the assistant's response to briefly address the interruption before seamlessly transitioning back to its original point (e.g., "A host cell is... Now, as I was saying..."). The remainder of the dialogue remains unchanged. This process yields a dataset enriched with more complex conversational flows, better preparing MoST for interactive scenarios.

### A.2.2 TEXT-TO-SPEECH INSTRUCTION DATASET CONVERSION

To enable our model with both speech and text instruction following abilities, we employ a strategy where we leverage the capabilities of our own MoST model, specifically after its initial post-training on ASR and TTS data (as described in Appendix A.1).

For a given text-based instruction sample (e.g., `{instruction: text_prompt, input: text_input, output: text_response}`), we use the ASR-TTS trained MoST model to synthesize the speech equivalent of the `text_response`. This converts the sample into a speech-output format (e.g., `{instruction: text_prompt, input: text_input, output: speech_response}`). Depending on the target task, we can also synthesize speech for the `text_input` or even the `instruction` itself, creating varied multimodal instruction formats. This approach allows us to efficiently repurpose vast quantities of high-quality text instructions for speech-based fine-tuning without the need for manual speech data collection, significantly broadening the range of tasks MoST is trained on.

### A.3 MIXED SPEECH-TEXT INSTRUCTION FINE-TUNING AND HYPER-PARAMETERS (STAGE 3)

Our training process for MoST (after Stage 1 post-training described in Appendix A.1 and Stage 2 data construction in Appendix A.2) proceeds to instruction fine-tuning.

**Instruction Fine-tuning Strategy.** This stage uses the curated dataset comprising synthesized interrupted dialogues and text-to-speech converted instructions. The goal is to teach the model complex reasoning, interaction, and command-following involving both modalities. Crucially, to mitigate catastrophic forgetting of the foundational ASR and TTS capabilities learned in Stage 1, the instruction fine-tuning data is mixed with a significant portion of the original ASR and TTS data. This ensures that MoST retains its core speech processing abilities while acquiring higher-level instruction-following competence. The mixing ratio between new instruction data and the ASR/TTS data is carefully managed throughout this stage.

## B TRAINING SETTINGS

This section details the hyper-parameters used for training MoST and the computing resources utilized for the experiments.

## B.1 TRAINING HYPER-PARAMETERS

Across all training stages (post-training and fine-tuning), we utilize the AdamW optimizer (Loshchilov & Hutter, 2019) with a cosine learning rate schedule, including a warm-up phase. Specific values for the peak learning rate, weight decay, batch size, and the exact number of training steps or epochs for each stage, along with the mixing ratios used during instruction fine-tuning are as follows:

Peak Learning Rate: 5e-5, Weight Decay: 0.01, Batch Size for ASR-TTS Post-training: 256, Batch Size for Mixed Instruction Following Fine-tuning: 128, Fine-tuning Steps: 10000, Warmup Steps: 1000, Mixing Ratio (Instruction:ASR/TTS): 1:1. These parameters were chosen based on preliminary experiments to ensure stable and efficient convergence.

## B.2 EXPERIMENT COMPUTING RESOURCES

The training experiments for MoST were conducted using a distributed setup comprising 48 NVIDIA A100 GPUs. Specifically, this involved 6 nodes, each equipped with 8 A100 GPUs.

## C MODEL CONFIGURATIONS

Table 4 details the model configurations used for MoST.

Table 4: Model configurations from `config_mostc.json`.

|  | Value |
| --- | --- |
|  | ["MoSTCForCausalLM"] |
| _bias | false |
| _dropout | 0.0 |
| _expert_indices | [32, 33, ..., 62, 63] |
| *auto_map*: |  |
|   AutoConfig | `configuration_MoST.MoSTCConfig` |
|   AutoModel | `modeling_most.MoSTCModel` |
|   AutoModelForCausalLM | `modeling_most.MoSTCForCausalLM` |
| _loss_alpha | 0.001 |
| _token_id | 100000 |
| _token_id | 100001 |
| _k_dense_replace | 1 |
| _act | silu |
| _size | 2048 |
| _range | 0.02 |

Table 4: Model configurations from `config_mostc.json` (Continued).

|                                    | Value   |
| ---------------------------------- | ------- |
| _size                              | 10944   |
| _lora_rank                         | 512     |
| _position_embeddings               | 163840  |
| _type                              | MoSTC   |
| _intermediate_size                 | 1408    |
| _layer_freq                        | 1       |
| _group                             | 1       |
| _routed_experts                    | 64      |
| _shared_experts                    | 2       |
| _topk_prob                         | false   |
| _attention_heads                   | 16      |
| _experts_per_tok                   | 6       |
| _hidden_layers                     | 27      |
| _key_value_heads                   | 16      |
| _tp                                | 1       |
| _lora_rank                         | null    |
| _nope_head_dim                     | 128     |
| _rope_head_dim                     | 64      |
| _norm_eps                          | 1e-06   |
| *rope_scaling*:                    |         |
|   beta_fast              | 32      |
|   beta_slow              | 1       |
|   factor                 | 40      |
|   mscale                 | 0.707   |
|   mscale_all_dim         | 0.707   |
|   original_max_position_embeddings | 4096 |

Table 4: Model configurations from `config_mostc.json` (Continued).

|  | Value |
| --- | --- |
| type | yarn |
| _theta | 10000 |
| _scaling_factor | 1.0 |
| _func | softmax |
| _aux | true |
| _expert_indices | [0, 1, ..., 30, 31] |
| _word_embeddings | false |
| _group | 1 |
| _method | greedy |
| _dtype | bfloat16 |
| _version | 4.33.1 |
| _cache | true |
| _modality_aware_routing | true |
| _head_dim | 128 |
| _size | 102400 |
| _continuous_audio | true |
| _model_path | /.../hubert_MoSTC/mhubert_base_25hz.pt |
| _model_path | /.../hubert_MoSTC/L11_quantizer_500.pt |
| _ckpt_type | converted |
| _hidden_size | 768 |
| _audio_token_id | 100502 |
| _audio_token_id | 100503 |
| _audio_wave_token_id | 100504 |
| _audio_wave_token_id | 100505 |

## D  LIMITATIONS, SOCIAL IMPACTS AND FUTURE WORK

MoST presents both potential societal benefits (e.g., accessibility) and risks (e.g., voice spoofing).

**Rationale for Index-Based Partition.** In standard sparse MoE models like DeepSeek-V2, experts are randomly initialized and do not exhibit semantic clustering (e.g., specialized "text experts" vs. "math experts") prior to training. Consequently, a random index-based split at initialization is statistically neutral—no particular subset of experts has inherent advantages for any specific modality. Our Stage 1 Cross-Modal Post-Training is explicitly designed to specialize these initially random experts into their assigned modalities through targeted ASR/TTS training.

**Role of Shared Experts.** Critically, the Shared Experts in MAMoE process every token regardless of modality, providing a mechanism to preserve general capabilities and facilitate cross-modal knowledge transfer. This architectural choice helps mitigate the risk of losing valuable knowledge during the hard partition, as evidenced by MoST's strong performance on text-based benchmarks (Appendix E).

**Future Directions.** While our approach demonstrates strong empirical results, we recognize that the index-based partition may not represent the theoretical optimum. More sophisticated initialization strategies represent important directions for future research, including: (1) *Clustering-based initialization*: Analyzing pretrained expert activations on representative data to group experts by their learned specializations before partitioning; (2) *Knowledge-preserving partitioning*: Developing methods to identify and preserve critical expert subsets for each modality; (3) *Adaptive expert allocation*: Dynamically adjusting the ratio of modality-specific experts during training based on task requirements.

Future work presents several exciting avenues. A primary direction is extending the MAMoE concept to incorporate additional modalities, particularly vision, to create a truly comprehensive audio-visual-language model. Further research could also explore scaling MoST to larger parameter counts, investigating different types of expert specializations, and evaluating its capabilities on a broader range of downstream speech and multimodal tasks.

## E  TEXT-BASED EVALUATIONS

To comprehensively assess MoST's capabilities beyond speech-specific tasks, we evaluate its performance on standard text-based benchmarks. Table 5 presents results across four diverse text benchmarks: MMLU (general knowledge), TriviaQA (factual question answering), GSM8K (mathematical reasoning), and HumanEval (code generation).

Table 5: Performance on Text-Based Evaluation Benchmarks. MoST demonstrates strong text understanding and generation capabilities, outperforming most speech-native models while maintaining competitive performance with text-focused systems. Higher scores are better. **MoST (Ours)** results are highlighted with green background.

| Model | MMLU | TriviaQA | GSM8K | HumanEval |
|---|---|---|---|---|
| MinMo | **58.5** | 65.8 | 58.1 | 42.3 |
| Llama-Omni2 | 44.7 | 35.2 | 21.8 | 13.2 |
| Moshi | 49.8 | 48.5 | 40.3 | 25.4 |
| Qwen2-Audio | 45.2 | 34.1 | 24.3 | 11.7 |
| SpiritLM | 36.9 | 42.0 | 21.5 | 9.7 |
| **MoST (Ours)** | 55.4 | **67.1** | **66.9** | **58.2** |

MoST significantly outperforms other open-source speech-native models (SpiritLM, Moshi, Qwen2-Audio) and recent end-to-end models like Llama-Omni2 across text tasks. MoST also maintains strong general reasoning capabilities (GSM8K 66.9) and coding abilities (HumanEval 58.2), confirming that our shared-expert design successfully mitigates catastrophic forgetting. While MinMo achieves a higher MMLU score (58.5 vs 55.4), MoST demonstrates superior performance on TriviaQA, GSM8K, and HumanEval, highlighting the effectiveness of our MAMoE architecture in preserving and enhancing text-based capabilities during multimodal training.

# F    USE OF LLMS

As this work is centered on large language models (LLMs), they are inherently part of our study. We clarify the roles of LLMs in four distinct but limited capacities:

- **Base model**. The core of our proposed MoST model is built upon an existing LLM as its backbone, consistent with the research objective of studying and advancing LLM-based methods. And we include it as part of reproducibility as well.

- **LM-guided synthesis.** In data synthesis section, LLMs were used as judges and converters to evaluate candidate outputs and perform format conversion. These roles were auxiliary, and MoST's effectiveness does not depend on LLM-specific heuristics.

- **Baselines.**  We adopted open-source LLMs as baselines to ensure fair and transparent comparisons, with results reproduced from official or widely accepted implementations.

- **Language correction.** LLMs were employed only for grammar correction and minor error detection in the manuscript, without altering the scientific content or analysis.

Overall, LLMs were employed only in auxiliary or benchmarking roles. Importantly, LLMs were never used for idea generation or the formulation of scientific arguments. All core algorithmic design, experimental findings, and analyses remain fully attributable to our proposed methodology.

