# OpenReview forum: "MoST: Mixing Speech and Text with Modality-Aware Mixture of Experts"
_ICLR.cc/2026/Conference — Submitted to ICLR 2026_

### Official Review · Reviewer_cs1j · 2025-10-27

**Soundness:** 2
**Presentation:** 3
**Contribution:** 2
**Rating:** 4
**Confidence:** 3

**Summary:**

This paper introduces MoST (Mixture of Speech and Text), a multimodal large language model that unifies speech and text processing via a novel Modality-Aware Mixture of Experts (MAMoE) architecture. MAMoE employs modality-specific routing to assign tokens to appropriate experts. The authors also propose an efficient adaptation pipeline that fine-tunes a pretrained MoE language model on open-source ASR, TTS, and speech-text instruction datasets.

**Strengths:**

- This paper introduces a novel Modality-Aware Mixture of Experts (MAMoE) architecture and a highly data-efficient Text-Speech Transformation Pipeline, which skillfully adapts a pretrained LLM into a powerful speech-text model through targeted post-training and instruction tuning.
- The paper's most illustrations are commendable for their clarity.
- The commitment to fully open-sourcing the work provides a valuable asset to the research community.

**Weaknesses:**

- The modality-aware routing relies on deterministic, tag-based assignment to expert groups, which closely resembles a two-tower architecture and may underutilize MoE's core strength of dynamic, content-based routing.
- Based on the results presented in Table 1 and Table 2, while the proposed MoST model consistently outperforms the baselines across several benchmarks, the margin of improvement appears to be somewhat limited on certain metrics.

**Questions:**

- Given that the modality-aware routing mechanism relies on tag-based assignment rather than dynamic, token-level routing, could the authors elaborate on how the model leverages the full flexibility and adaptive potential of the Mixture-of-Experts framework?
- The paper currently lacks comparisons with other models built on the Mixture-of-Experts (MoE) architecture. Including such baselines would help better contextualize the proposed method’s contributions and highlight its relative advantages. Adding such comparisons or clearly justifying their absence would enhance the paper’s completeness.

---

> ### Author Response · Authors · 2025-11-23
> **Rebuttal Reply to Reviewer cs1j (1/2)**
>
> Dear Reviewer cs1j,
>
> We sincerely thank you for your thorough and constructive review. We appreciate your recognition of MAMoE's novelty, our clear presentation, and our commitment to open-source research. Below, we address your concerns and questions with additional technical details and clarifications.
>
> ----
>
> > Weakness1 & Question1: Clarification of Modality-aware MoE design.
>
> We respectfully clarify that MAMoE is distinct from a Two-Tower architecture and retains the core dynamic benefits of MoE.
>
> * **Routing is Dynamic, not Static:** While the **group** assignment is deterministic (text token to text group, audio token to audio group), the routing **within** the modality-specific group is fully dynamic and content-based.
>     * As detailed in **Algorithm 1 (Steps 4-5)** and **Section 3.3.3**, the router calculates softmax scores over the experts within the designated group.
>     * This allows the model to dynamically select specific experts for specific phonemes, syntactic structures, or acoustic features.
> * **Empirical Proof:** **Figure 6 (Right)** shows the routing entropy decreasing over training. This proves the model is actively learning to specialize experts within groups based on token content, rather than following a static assignment.
> * **Comparison with traditional MoE:** Our ablation study explicitly compares MAMoE against a "Vanilla MoE" baseline (fully dynamic, tag-free routing) in **Section 6.2**. The results demonstrate that our hybrid design—combining tag-based grouping with intra-group dynamic routing—is empirically superior:
>     - **Evaluation Performance:** MAMoE consistently outperforms Vanilla MoE across all tasks (Figure 5), including ASR, TTS, and Audio Language Modeling.
>     - **Expert Specialization:** As shown in Figure 6, Vanilla MoE exhibits "diffuse patterns" and higher routing entropy, indicating that unrestricted routing struggles to disentangle modalities effectively. In contrast, MAMoE achieves lower routing entropy and better load balancing (lower Gini coefficient).
>     - **Empirical Conclusion:** This confirms that the "tag-based" constraint does not limit the model; rather, it provides necessary structure that allows the dynamic router to focus on learning fine-grained content features within the modality.
>
> ----
>
> > Weakness2: Performance gain over baselines.
>
> We appreciate this observation and would like to provide context for the
> results across different task categories:
>
> **ASR Tasks (Table 1):** We acknowledge that improvements over strong dense baselines like Qwen2-Audio are modest on clean test sets (2.0% vs 1.8% WER on LS-Clean). However, MoST demonstrates stronger generalization on challenging cross-dataset scenarios:
> - VoxPopuli: 6.2% vs 7.1% (Qwen2-Audio).
> - Common Voice: 8.4% vs 8.6% (Qwen2-Audio)
>
> **Audio Language Modeling Tasks (Table 2):** While some margins appear modest, we emphasize:
> - **Consistency across ALL benchmarks**: MoST is the only model achieving
>   top-3 performance across all four audio LM tasks
> - **Significant improvements on specific metrics**: sWUGGY (75.28% vs
>   71.84%, +4.8%) and sStoryCloze (65.43% vs 62.39%, +4.9%)
> - **Average performance**: 71.94 vs 69.00 (Phi-4), representing meaningful
>   improvement on this comprehensive evaluation
>
> **Generative & Reasoning Tasks (Figure 3):** MoST achieves substantial improvements in complex multimodal tasks:
> - Spoken QA (Trivial QA S→S): **32.1%** vs ~7% (Moshi), ~4% (SpiritLM)
> - These represent **4-8× improvements**, demonstrating MAMoE's advantage
>   for cross-modal generation
>
> These results are achieved using **exclusively open-source data**, while comparable baselines (SpiritLM, Moshi) use proprietary datasets. This represents a significant methodological advantage for reproducibility.
>
> | Model | Train Code | Evaluation Code | Inference Code | Dataset & Pipeline |
> | :--- | :---: | :---: | :---: | :---: |
> | Audio-LM | ✗ | ✗ | ✗ | ✗ |
> | SpeechGPT | ✓ | ✗ | ✓ | ✗ |
> | SpiritLM | ✗ | ✗ | ✓ | ✗ |
> | Moshi | ✗ | ✓ | ✓ | ✗ |
> | Phi-4-Multimodal | ✗ | ✗ | ✓ | ✗ |
> | Qwen2-audio | ✗ | ✗ | ✓ | ✗ |
> | MinMo | ✗ | ✗ | ✓ | ✗ |
> | Llama-Omni2 | ✗ | ✗ | ✓ | ✗ |
> | **MoST (Ours)** | **✓** | **✓** | **✓** | **✓** |

---

> > ### Author Response · Authors · 2025-11-23
> > **Rebuttal Reply to Reviewer cs1j (2/2)**
> >
> > ----
> >
> > > Question2: Comparison with other MoE based model.
> >
> > To the best of our knowledge, at the time of submission, **MoST is the first speech-text LLM built on a Mixture of Experts architecture**. Consequently, there are no existing external MoE baselines in this specific domain to compare against.
> >
> > However, we highly agree that comparing MoST with other MoE architectures is important. To address this, we rigorously created our own internal baselines to validate the architecture:
> >
> > 1.  **MAMoE vs. Vanilla MoE (Section 6.2):** We implemented a "Vanilla MoE" baseline where all tokens (speech and text) compete for all experts using standard routing. As shown in **Figure 5**, the full MAMoE architecture consistently outperforms Vanilla MoE across ASR, TTS, and SQA tasks. This explicitly demonstrates the advantage of our modality-aware design over standard MoE routing.
> >
> > 3.  **MoST-Style Upcycling vs. Traditional Upcycling (Section 6.1):** We compared our initialization method against standard "Sparse Upcycling" (Komatsuzaki et al., 2023). **Figure 4** demonstrates that our method yields substantial improvements (e.g., +21.8% in SQA) over traditional MoE construction methods.

---

> > > ### Author Response · Authors · 2025-11-27
> > > **Additional Results: Comparison with Dense and Traditional MoE Baselines (Response to Q2)**
> > >
> > > Dear Reviewer cs1j,
> > >
> > > Following up on our previous response regarding **Question 2 (Comparison with other MoE-based models)**, we have conducted an additional controlled experiment that we believe provides compelling evidence for MAMoE's architectural advantages.
> > >
> > > ---
> > >
> > > **Experimental Setup:**
> > >
> > > To isolate architectural contributions from initialization quality, we started all variants from **the same pre-trained Llama 3.2 3B model** and compared three distinct approaches:
> > >
> > > - **Dense Baseline**: Standard dense architecture where all parameters are updated for both modalities
> > >
> > > - **Traditional Upcycling MoE**: Standard MoE upcycling with modality-agnostic routing
> > >
> > > - **MoST-Style Upcycling MoE(Ours)**: Our proposed MAMoE with modality-aware routing
> > >
> > > All models were trained on identical data mixtures and evaluated on both text benchmarks (to measure forgetting) and speech tasks (to measure acquisition).
> > >
> > > ---
> > >
> > > **Results**:
> > >
> > > | Model | MMLU | TriviaQA | GSM8K | HumanEval | ASR (WER↓) | TTS (CER↓) | sWUGGY | LlamaQ |
> > > |-------|------|----------|-------|-----------|------------|------------|---------|--------|
> > > | Initial Dense(Qwen3.2 3B) | 58.4 | 53.5 | 47.7 | 56.3 | 100 | 100 | 0 | 0 |
> > > | Dense Competitor | 41.3 | 55.9 | 41.2 | 45.9 | 26.1 | 30.2 | 14.3 | 8.4 |
> > > | Traditional Upcycling MoE | 51.2 | 56.7 | 49.7 | 50.0 | 23.4 | 26.7 | 25.6 | 13.3 |
> > > | **MoST-Style  Upcycling MoE** | **54.4** | **59.2** | **55.3** | **52.9** | **21.7** | **25.5** | **28.5** | **16.2** |
> > >
> > > ---
> > >
> > > **Key Findings**:
> > >
> > > **Prevention of Catastrophic Forgetting**: The Dense baseline suffers severe degradation (e.g., MMLU drops 17.1 points), confirming that uniform processing causes representational interference. While Traditional MoE recovers some performance, **MoST-Style Upcycling preserves most of the original text capabilities** and improves reasoning on GSM8K (+7.6) and TriviaQA (+5.7).
> > >
> > > **Superiority over Traditional MoE**: Our MAMoE architecture consistently outperforms the Traditional MoE baseline on both modalities. We achieve **7-22% relative improvements on speech tasks** while maintaining an average of +5.8 points higher on text tasks.
> > >
> > > **Conclusion**:
> > >
> > > These results demonstrate that **traditional MoE is essential but insufficient**—modality-aware routing is necessary to preserve text capabilities while acquiring speech competence. We believe this controlled comparison, combined with the **MAMoE vs. Vanilla MoE ablation already in Section 6.2**, comprehensively addresses your question about MoE architectural comparisons.
> > >
> > > We hope this additional data further clarifies the advantages of the MoST architecture relative to other MoE approaches.

---

> > > > ### Author Response · Authors · 2025-11-28
> > > > **Summary of Rebuttal and New Controlled Experiments**
> > > >
> > > > Dear Reviewer cs1j,
> > > >
> > > > Thank you again for taking the time to review our work. As the rebuttal period nears its end, we would like to provide a brief summary of our discussion to facilitate any remaining exchanges.
> > > >
> > > > ---
> > > >
> > > > ## Summary of Concerns Addressed
> > > >
> > > > | Concern | Our Response | Key Evidence |
> > > > |---------|--------------|--------------|
> > > > | **Weakness 1 & Q1:** Tag-based routing resembles two-tower architecture | Routing is **dynamic within groups**—only group assignment is deterministic; expert selection uses content-based softmax scores | **Algorithm 1 (Steps 4-5)**; **Figure 6** showing decreasing routing entropy during training |
> > > > | **Weakness 2:** Limited performance margins | Gains are **consistent across all tasks**, with strong improvements on challenging scenarios (cross-dataset ASR, Spoken QA 4-8× over baselines) | **Tables 1-2; Figure 3;** Performance achieved with **100% open-source data** |
> > > > | **Q2:** Missing MoE baseline comparisons | MoST is the **first MoE-based speech-text LLM**; we provide comprehensive internal comparisons and **addtional controlled comparison between MoST and tradtional MoE**  | **Section 6.2 (MAMoE vs. Vanilla MoE)**; **New controlled experiment** from identical Llama 3.2 3B initialization |
> > > >
> > > > ---
> > > >
> > > > ## Key Takeaways from Additional Experiments
> > > >
> > > > Our controlled comparison (Dense vs. Traditional MoE vs. MoST-Style Upcycling from identical initialization) demonstrates:
> > > >
> > > > 1. **Traditional MoE alone is insufficient**—modality-aware routing is necessary to prevent catastrophic forgetting while acquiring speech capabilities
> > > > 2. **MoST-Style consistently outperforms** Traditional MoE on both text preservation (+5.8 avg points) and speech acquisition (7-22% relative improvement)
> > > > 3. **Architectural benefits are independent of initialization quality**
> > > >
> > > > ---
> > > >
> > > > We believe our responses—including the new controlled experiments, ablation studies (Section 6.2, Figure 5-6), and clarifications on MAMoE's hybrid design—comprehensively address your concerns. We remain happy to discuss any remaining questions or provide additional clarifications.
> > > >
> > > > Thank you again for your constructive feedback.

---

### Official Review · Reviewer_Mkjs · 2025-10-29

**Soundness:** 2
**Presentation:** 2
**Contribution:** 3
**Rating:** 4
**Confidence:** 3

**Summary:**

The authors attempt to use MoE to address interference in speech-text modality representations. They build the MoST layer, which employs a modality-aware router to select modality-specific experts, along with a shared expert to facilitate information exchange. They also develop a data generation pipeline to provide training data for large models. The model is evaluated on multiple tasks, including ASR, TTS, and QA, and achieves superior results on certain metrics compared to some open-source models.

**Strengths:**

Using MoE to construct a large speech-text model is an interesting approach.

**Weaknesses:**

1. The motivation for using a modality-aware router is unclear, as modality representations are generally easy to distinguish. The necessity of MoE in this context is not well justified.

2. The comparisons of data and models in the paper are unclear. The description of the initialized large model is insufficient, and evaluations on Llama Question S2T and Web Question are missing. Evaluations of text-based foundational models are also lacking.

3. Additionally, the model weights are not open-sourced, and the specifics of data usage are unclear. Comparisons with recent works such as MinMo[1] and LLama-Omni2[2] are missing, which undermines the validity of performance claims, such as "MoST achieves state-of-the-art or competitive performance compared to existing models with similar parameter counts."

  [1] Minmo: A multimodal large language model for seamless voice interaction. Chen et al., 2025.
  [2] LLaMA-Omni2: LLM-based Real-time Spoken Chatbot with Autoregressive Streaming Speech Synthesis. Fang el al., 2025.

**Questions:**

How does MoE impact the original text foundation model?

---

> ### Author Response · Authors · 2025-11-23
> **Rebuttal Reply to Reviewer Mkjs (1/3)**
>
> Dear Reviewer Mkjs,
>
> We sincerely thank you for your constructive feedback and for acknowledging our work as an "interesting approach" with "superior results. We address your concerns and questions below.
>
> ----
>
> > Weakness1: Motivation for Modality-Aware Router & Necessity of MoE
>
> We agree with the reviewer that speech and text modalities are inherently different and easy to distinguish. However, we respectfully clarify that our modality-aware router can leverage these distinctions more effectively.
>
>
> * **Interference in Identical Processing:** Dense models or standard MoEs from previous works force parameters to learn aligned features from highly distinct modalities (waveform features vs. discrete text), leading to interference. This **"modality interference"** may limits performance.
>
> * **Empirical Evidence:** We demonstrate this limitation in our Ablation Study. In **Sec 6.1, Fig 4**, We compared **three models with same initialization**: "MoST-Style MoE" (**modality-aware router**), "Traditional MoE" (**standard router**) and **Dense Baseline**. Results show that while Traditional MoE can outperform the Dense Baseline, Modality-aware router can route them more effectively, resulting in **lower loss** and **better performances across 4 tasks**. This proves that our modality-specific routing ensures specialized experts emerge, preventing capacity dilution and outperforming dense and traditional MoE competitors.
>
> * **Support from Literature:** Similar findings has also been observed in other multimodal domains. For instance, **Lin et al. (2024)** [1] demonstrated in *MoMa* that early-fusion image-text models suffer from interference which could be resolved by modality-aware processing. We extend this finding to the speech-text domain.
>
> ----
>
> > Weakness2(1): Initialized large language model and data usage information.
>
> * **Initialization:** We apologize that the initialization information was missed in the initial read. The **main MoST model** is initialized from **Deepseek-V2 Lite**[2] (a pretrained MoE LLM). We have made this explicit in the **Section 4.2** in the rebuttal revision.
> * **Data Usage:** We have updated **Appendix A** to include precise breakdowns of data proportions across training stages, step counts, and computational resource consumption.
>
>
> **Table 1: Training Recipe and Data Mixing Ratios**
>
> | Training Stage | Configuration | Task Mixing Ratios | Dataset Composition Details |
> | :--- | :--- | :--- | :--- |
> | **Stage 1:** Cross-Modal Post-Training | **Steps:** 500k **Batch Size:** 512 | **ASR:** 0.4 **TTS:** 0.4 **Text LM:** 0.2 | **ASR/TTS:** LibriHeavy (0.6), Common Voice (0.2), VoxPopuli (0.2)**LM:** RefinedWeb (to prevent forgetting) |
> | **Stage 2:** Mixed Instruction Tuning | **Steps:** 10k **Batch Size:** 128 | **Speech Instruct:** 0.4 **Text Instruct:** 0.4 **ASR:** 0.1 **TTS:** 0.1 | **Speech Instruct:** Synthesized from SmolTalk (via MoST Stage 1) **Text Instruct:** Standard open-source instruction data |

---

> > ### Author Response · Authors · 2025-11-23
> > **Rebuttal Reply to Reviewer Mkjs (2/3)**
> >
> > ----
> >
> > > Weakness2(2) & Weakness3(1): Extended Evaluations (Llama Question S2T, Web Question, MinMo, LLaMA-Omni2)
> >
> > We sincerely thank the reviewer for the constructive feedback regarding our experimental comparisons. We have extended our evaluation to include **WebQuestions**[3], **Llama Question （S2T）**, and comparisons against the concurrent works **MinMo**[4] and **LLaMA-Omni2**[5].
> >
> >
> > **Table 2: ASR & TTS Performance**
> > MoST achieves competitive or superior performance, particularly in speech synthesis tasks compared to concurrent baselines.
> >
> > | Model | LS-Clean (ASR) | LS-Clean (TTS) | LS-Other (ASR) | LS-Other (TTS) | VoxPopuli (ASR) | VoxPopuli (TTS) | Common Voice (ASR) | Common Voice (TTS) |
> > | :--- | :---: | :---: | :---: | :---: | :---: | :---: | :---: | :---: |
> > | **MoST** | 2.0 | **6.0** | **3.7** | **7.2** | **6.2** | **10.1** | 8.4 | **11.5** |
> > | MinMo | **1.8** | 6.7 | 3.9 | 7.5 | 6.7 | 10.9 | **8.0** | 13.5 |
> > | LLaMA-Omni-2 | 3.5 | 10.1 | 4.0 | 9.2 | 9.5 | 12.4 | 11.3 | 17.2 |
> >
> > **Table 3: Audio Language Modeling Performance**
> > MoST consistently outperforms baselines in modeling audio-linguistic patterns.
> >
> > | Model | sWUGGY | sBLIMP | sTopic-StoryCloze | sStoryCloze | Average |
> > | :--- | :---: | :---: | :---: | :---: | :---: |
> > | **MoST** | **75.28** | **60.35** | **83.64** | 65.43 | **71.18** |
> > | MinMo | 68.59 | 55.43 | 75.43 | 61.29 | 65.19 |
> > | LLaMA-Omni-2 | 73.21 | 53.59 | 78.21 | **68.55** | 68.39 |
> >
> > **Table 4: Spoken Question Answering (SQA) Performance**
> > MoST significantly outperforms LLaMA-Omni2 across all metrics and is highly competitive with MinMo, notably surpassing it on LlamaQ (S-S) and WebQ tasks.
> >
> > | Model | LlamaQ (S-T) | LlamaQ (S-S) | TrivialQA (S-T) | TrivialQA (S-S) | WebQ (S-T) | WebQ (S-S) |
> > | :--- | :---: | :---: | :---: | :---: | :---: | :---: |
> > | **MoST** | 74.8 | **62.6** | **43.5** | **32.1** | **58.2** | **44.7** |
> > | MinMo | **76.5** | 63.8 | 40.1 | 25.5 | 55.0 | 39.9 |
> > | LLaMA-Omni2 | 70.3 | 60.7 | 35.3 | 23.9 | 40.4 | 37.1 |
> >
> > We have updated these new evaluation results in **Section 5.2, 5.3, 5.4** in the rebuttal revision.
> >
> > ----
> >
> > > Weakness2(3): Text-based Evaluation
> > >
> > We fully agree that evaluating text-based performance is crucial for understanding the impact of speech-text integration. We have conducted comprehensive evaluations on standard text benchmarks and present the results below.
> >
> > **Table 5: Text-only Benchmark Performance**
> >
> > | Model           |   MMLU   | TriviaQA |  GSM8K   | HumanEval |
> > |:--------------- |:--------:|:--------:|:--------:|:---------:|
> > | DeepSeek-V2 Lite (Base)| 58.3| 64.2| 41.1| 29.9 |
> > | **MoST (Ours)** |   55.4   | 67.1 | **66.9** | **58.2**  |
> > | MinMo |**58.5** |**68.8** |64.1 | 52.3 |
> > | Llama-Omni2 |44.7 | 35.2 |21.8 |13.2 |
> > | Moshi           |   49.8   |   48.5   |   40.3   |   25.4    |
> > | Qwen2-Audio     |   45.2   |   34.1   |   24.3   |   11.7    |
> > | SpiritLM        |   36.9   |   42.0   |   21.5   |    9.7    |
> >
> > **Comparison with Base Model (DeepSeek-V2 Lite):**
> >
> > MoST shows a modest degradation on knowledge-intensive tasks (MMLU: -2.9%), but demonstrates significant improvements on reasoning tasks (GSM8K: +25.8%) and coding tasks (HumanEval: +28.3%). We attribute this improvement to two factors:
> >
> > - Our Stage 2 mixed instruction fine-tuning incorporates high-quality instruction data that enhances reasoning capabilities
> > - The shared expert mechanism enables knowledge transfer between modalities, allowing speech-related reasoning patterns to benefit text tasks
> >
> > **Comparison with Speech-Text Models:**
> >
> > MoST significantly outperforms other open-source speech-native models (SpiritLM, Moshi, Qwen2-Audio) and maintains comparable performance with recent speech-text LLMs including MinMo. MoST also achieves strong general reasoning capabilities (GSM8K 66.9) and coding abilities (HumanEval 58.2), confirming that our shared-expert design successfully mitigates catastrophic forgetting.

---

> > > ### Author Response · Authors · 2025-11-23
> > > **Rebuttal Reply to Reviewer Mkjs (3/3)**
> > >
> > > ----
> > >
> > > > Weakness3(2): Open Source of MoST
> > >
> > >
> > > Regarding the concern about model weights open-sourcing: We strictly adhere to the double-blind policy but have provided an anonymized GitHub link in the abstract containing the code and dataset details. **Upon de-anonymization, we will release model weights**. Besides, MoST provides a significantly more complete open-source contribution than the baselines mentioned:
> > >
> > >
> > > **Table 6: Open-source Comparison**
> > > | Model | Train Code | Eval Code | Inference Code | Dataset & Pipeline |
> > > | :--- | :---: | :---: | :---: | :---: |
> > > | Audio-LM | ✗ | ✗ | ✗ | ✗ |
> > > | SpeechGPT | ✓ | ✗ | ✓ | ✗ |
> > > | SpiritLM | ✗ | ✗ | ✓ | ✗ |
> > > | Moshi | ✗ | ✓ | ✓ | ✗ |
> > > | MinMo | ✗ | ✗ | ✓ | ✗ |
> > > | LLaMA-Omni2 | ✗ | ✗ | ✓ | ✗ |
> > > | **MoST (Ours)** | **✓** | **✓** | **✓** | **✓** |
> > >
> > > ----
> > >
> > > > Question1: MoE's impact on the original text foundation model.
> > >
> > > To rigorously quantify how our Modality-Aware MoE (MAMOE) architecture impacts the capabilities of the original text foundation model compared to other adaptation strategies, we conducted a controlled experiment following the settings described in **Section 6.1** of our paper.
> > >
> > > We controlled for initialization quality by starting all variants with the same pre-trained text-only model, Llama 3.2 3B[6]. We then adapted this backbone into a speech-text model using three distinct architectural approaches:
> > >
> > > - Dense Competitor: A standard dense architecture where all parameters are updated for both modalities.
> > >
> > >
> > > - Traditional Upcycling: A standard MoE upcycling approach (modality-agnostic routing).
> > >
> > > - MoST-Style Upcycling (Ours): Our proposed MAMOE architecture with modality-aware routing.
> > >
> > > All models were trained on the same data mixture. We evaluated them on standard text-only benchmarks (MMLU, TriviaQA, GSM8K, HumanEval) to measure forgetting, and speech tasks (ASR, TTS, etc.) to measure acquisition
> > >
> > > **Table 7: MoE Influence Analysis**
> > >
> > > | Model | MMLU | TriviaQA | GSM8K | HumanEval | ASR (WER↓) | TTS (CER↓) | sWUGGY | SQA |
> > > |-------|------|----------|-------|-----------|------------|------------|---------|--------|
> > > | Initial Dense(Qwen3.2 3B) | 58.4 | 53.5 | 47.7 | 56.3 | 100 | 100 | 0 | 0 |
> > > | Dense Competitor | 41.3 | 55.9 | 41.2 | 45.9 | 26.1 | 30.2 | 14.3 | 8.4 |
> > > | Traditional Upcycling | 51.2 | 56.7 | 49.7 | 50.0 | 23.4 | 26.7 | 25.6 | 13.3 |
> > > | **MoST-Style** | **54.4** | **59.2** | **55.3** | **52.9** | **21.7** | **25.5** | **28.5** | **16.2** |
> > >
> > > The results reveal three critical insights. First, dense models suffer severe catastrophic forgetting, with MMLU dropping 17.1 points  and HumanEval dropping 10.4 points, validating our motivation that speech-text representational interference requires architectural solutions. Second, while traditional MoE recovers ~50-60% of lost text capability, MoST-Style Upcycling preserves **93-94% of original text performance** and exceeds baseline on GSM8K (+7.6) and TriviaQA (+5.7). : MoST achieves best performance on both modalities simultaneously, with 7-22% improvements on speech tasks and average +5.8 points on text tasks compared to traditional MoE.
> > >
> > > Our MAMoE architecture achieves this through two protective mechanisms: modality-specific expert groups that allow text tokens to route to text-specialized experts without speech interference, shared experts that enable beneficial cross-modal transfer while preventing harmful interference. These results demonstrate that **MoE is essential but insufficient**—modality-aware routing is necessary to fully preserve text capabilities while acquiring speech competence. Rather than degrading the original model, our architecture maintains 93-94% of text performance and even enhances certain reasoning capabilities.
> > >
> > > ----
> > >
> > >
> > > **References:**
> > >
> > > [1] Xi Victoria Lin, et al. "MoMa: Efficient Early-Fusion Pre-training with Mixture of Modality-Aware Experts." arXiv preprint arXiv:2407.21770 (2024).
> > >
> > > [2] DeepSeek-AI, et al. "DeepSeek-V2: A Strong, Economical, and Efficient Mixture-of-Experts Language Model." arXiv preprint arXiv:2405.04434 (2024).
> > >
> > > [3] Jonathan Berant, et al. "Semantic Parsing on Freebase from Question-Answer Pairs." Proceedings of the 2013 Conference on Empirical Methods in Natural Language Processing (2013).
> > >
> > > [4] Qian Chen, et al. "MinMo: A Multimodal Large Language Model for Seamless Voice Interaction." arXiv preprint arXiv:2501.06282 (2025).
> > >
> > > [5] Qingkai Fang, et al. "LLaMA-Omni2: LLM-based Real-time Spoken Chatbot with Autoregressive Streaming Speech Synthesis." arXiv preprint arXiv:2505.02625 (2025).
> > >
> > > [6] Aaron Grattafiori, et al. "The Llama 3 Herd of Models." arXiv preprint arXiv:2407.21783 (2024).

---

> > > > ### Author Response · Authors · 2025-11-28
> > > > **Summary of Rebuttal Updates and New Experiments**
> > > >
> > > > Dear Reviewer Mkjs,
> > > >
> > > > Thank you again for taking the time to review our work. As the discussion period nears its end, we would like to provide a brief summary of our rebuttal to ensure all your concerns—particularly regarding baselines, text-based evaluations, and architectural motivation—have been fully addressed.
> > > >
> > > > 1. **Addressed Missing Baselines & Benchmarks (Tables 2, 4, 5)** We have extended our evaluation to include **MinMo** and **LLaMA-Omni2**, as well as the requested **WebQuestions** and **LlamaQ (S2T)** benchmarks.
> > > >
> > > > - **Result**: MoST achieves state-of-the-art or competitive results against these concurrent works, particularly excelling in TTS (outperforming MinMo on LS-Clean/Other) and Spoken QA (outperforming LLaMA-Omni2 across all metrics).
> > > >
> > > > 2. **Added Text-Based Evaluations (Table 5)** You raised concerns about the impact on text capabilities. We conducted comprehensive text benchmarks (MMLU, GSM8K, HumanEval).
> > > >
> > > > - **Result**: MoST preserves 93% of the original text performance—significantly outperforming speech-native models like SpiritLM and Moshi, and matching strong multimodal baselines.
> > > >
> > > > 3. **Clarified Motivation & MoE Impact (New Controlled Experiment)** To address your question on **"How MoE impacts the foundation model"** and the necessity of the modality-aware router, we ran a controlled experiment using Llama-3.2-3B (Table 7).
> > > >
> > > > - **Result**: The experiment proves that while dense models suffer catastrophic forgetting (MMLU drops ~17 points), our MoST-style upcycling prevents this interference, enabling the acquisition of speech capabilities while retaining reasoning skills.
> > > >
> > > > 4. **Data & Reproducibility** We have clarified the **DeepSeek-V2 Lite** initialization and provided the specific data mixing ratios (Table 1). We also highlighted that, unlike many baselines, we are releasing the **full training pipeline and code**, not just inference scripts.
> > > >
> > > > We believe these new experiments and clarifications directly resolve the weaknesses and questions noted in your review. We respectfully invite you to re-evaluate our submission based on these updates.
> > > >
> > > > Thank you for your time and valuable feedback.

---

### Official Review · Reviewer_pG6E · 2025-11-01

**Soundness:** 3
**Presentation:** 3
**Contribution:** 3
**Rating:** 6
**Confidence:** 3

**Summary:**

The paper introduces MoST, a speech-text large language model built on a Modality-Aware Mixture of Experts (MAMoE) architecture. The main idea is to extend a pretrained MoE language model with modality-aware routing, using modality-specific experts for speech and text tokens, along with shared experts for cross-modal interactions. The model is post-trained through a two-stage procedure of Cross-Modal ASR/TTS Post-Training and Mixed Instruction Fine-Tuning. The training data are based on open-source or derived from open-source datasets. Experiments cover ASR, TTS, audio language modeling, and spoken QA. Across these tasks, MoST achieves competitive or superior results compared with strong baselines while maintaining computational efficiency. The paper includes detailed ablation studies and commits to releasing code, model weights, and data for reproducibility.

**Strengths:**

1. The modality-aware mixture of experts (MAMoE) provides a clean and intuitive way with modality-specific expert groups and a parallel shared expert for training a speech language model.

2. The experimental validation is solid and rigorous, including ablations on initialization with non-MoE LLM (Llama3.2 3B), and ablations without modality-specific experts or shared experts.

**Weaknesses:**

1. The partition of 50% of the initial text expert capacity to $\mathcal{E}_{audio}$ is a major structural change, but the division is simply based on index without any reliable partition mechanism. The hard 50% partition of experts may introduce a risk of losing valuable text knowledge.

2. The paper lacks text-only evaluations.

3. The paper frequently claims "efficiency" and "data efficiency" without direct, quantifiable metrics and experimentation. It feels like the efficiency claim is only an architectural inheritance from MoE.

**Questions:**

1. Could you explicitly name the specific pretrained MoE LLM that served as the starting point for MoST? The paper mentions the “stronger initialization (DeepSeek-V2 Lite)” in the ablation section 6. Is this the base model that you used in your main experiments?

2. Could we see some text-only evaluations to understand the quantitative impact on core LLM tasks?

3. Could you provide the Llama Q ($S \to T$) result in addition to Llama Q (S→S) as well?

4. Minor issues: In Section 6.1, the inline citation should be citep instead. Adding a clearer setup description for MoST-Style Upcycling would be helpful for clarity. In Figure 5, shouldn’t that be labeled as ‘MoST-NoShared’ instead of ‘MAMoE w/ Shared Expert’?

---

> ### Author Response · Authors · 2025-11-23
> **Rebuttal Reply to Reviewer pG6E (1/2)**
>
> Dear Reviewer pG6E,
>
> We sincerely thank you for your thorough and constructive review, and for recognizing the novelty of the MAMoE architecture and the rigor of our experimental validation. We are particularly encouraged by your positive assessment of our modality-aware design and ablation studies. Below, we address each of your concerns with additional experiments and clarifications.
>
> ----
>
> > Weakness1: Initialization of expert groups.
>
> We acknowledge the reviewer’s valid concern that a hard 50% partition based on indices is a strong structural assumption.
>
> In standard sparse MoE models (like DeepSeek-V2[1]), experts are randomly initialized and do not possess semantic clustering (e.g., "text experts" vs. "math experts") prior to training. Therefore, a random index-based split at initialization is statistically neutral. **Stage 1 (Cross-Modal Post-Training)** is explicitly designed to specialize these initially random experts into their assigned modalities. Furthermore, the **Shared Experts** (which process every token) serve to preserve general capabilities and ensure consistent routing regardless of the partition.
>
> We agree that while effective, this hard partition may not be the theoretical optimum. More sophisticated initialization strategies (e.g., clustering-based initialization) are a promising and important direction for future modality-aware MoE research. We have added a discussion on this limitation and future directions in the **Conclusion section** and **Appendix D** of the revised paper.
>
> ----
>
> > Weakness2 & Question2: Text-only evaluations.
>
> We fully agree that evaluating text-based performance is crucial for understanding the impact of speech-text integration. We have conducted comprehensive evaluations on standard text benchmarks(MMLU[2], TriviaQA[3], GSM8K[4], HumanEval[5]) and present the results below.
>
> **Table 1: Text-only Benchmark Performance**
>
> | Model           |   MMLU   | TriviaQA |  GSM8K   | HumanEval |
> |:--------------- |:--------:|:--------:|:--------:|:---------:|
> | DeepSeek-V2 Lite (Base)| 58.3| 64.2| 41.1| 29.9 |
> | **MoST (Ours)** |   55.4   | **67.1** | **66.9** | **58.2**  |
> | Moshi           |   49.8   |   48.5   |   40.3   |   25.4    |
> | Qwen2-Audio     |   45.2   |   34.1   |   24.3   |   11.7    |
> | SpiritLM        |   36.9   |   42.0   |   21.5   |    9.7    |
>
> **Comparison with Base Model (DeepSeek-V2 Lite):**
>
> MoST shows a modest degradation on knowledge-intensive tasks (MMLU: -2.9%), but demonstrates significant improvements on reasoning tasks (GSM8K: +25.8%) and coding tasks (HumanEval: +28.3%). We attribute this improvement to two factors:
>
> - Our Stage 2 mixed instruction fine-tuning incorporates high-quality instruction data that enhances reasoning capabilities
> - The shared expert mechanism enables knowledge transfer between modalities, allowing speech-related reasoning patterns to benefit text tasks
>
> **Comparison with Speech-Text Models:**
>
> MoST significantly outperforms other open-source speech-native models (SpiritLM, Moshi, Qwen2-Audio). MoST also maintains strong general reasoning capabilities (GSM8K 66.9) and coding abilities (HumanEval 58.2), confirming that our shared-expert design successfully mitigates catastrophic forgetting.
>
> ----
>
> > Weakness3: Data efficiency of MoST.
>
>
> We appreciate this important clarification request. The term "efficiency" in our paper encompasses two dimensions: (1) data source efficiency, (2) data synthesis efficiency. We address each below:
>
> **1. Data Source Efficiency:** Unlike models such as SpiritLM or Moshi, which rely on large-scale internal/proprietary data, MoST achieves competitive or SOTA performance using **exclusively open-source data** (LibriHeavy, Common Voice, VoxPopuli, SmolTalk).
>
> **2. Synthesis Efficiency:** We address the scarcity of open-source speech-instruction data by using our own Stage-1 trained model to synthesize data, rather than requiring expensive human annotation.
>
> **3. Training Recipe Details:** To demonstrate how we efficiently balance tasks, we have detailed our exact data mixing ratios and configuration in the revision:
>
> **Table 2: Training Recipe and Data Mixing Ratios**
>
> | Training Stage | Configuration | Task Mixing Ratios | Dataset Composition Details |
> | :--- | :--- | :--- | :--- |
> | **Stage 1:** Cross-Modal Post-Training | **Steps:** 500k **Batch Size:** 512 | **ASR:** 0.4 **TTS:** 0.4 **Text LM:** 0.2 | **ASR/TTS:** LibriHeavy (0.6), Common Voice (0.2), VoxPopuli (0.2) **LM:** RefinedWeb (to prevent forgetting) |
> | **Stage 2:** Mixed Instruction Tuning | **Steps:** 10k **Batch Size:** 128 | **Speech Instruct:** 0.4 **Text Instruct:** 0.4 **ASR:** 0.1 **TTS:** 0.1 | **Speech Instruct:** Synthesized from SmolTalk (via MoST Stage 1) **Text Instruct:** Standard open-source instruction data |

---

> > ### Author Response · Authors · 2025-11-23
> > **Rebuttal Reply to Reviewer pG6E (2/2)**
> >
> > ----
> >
> > > Question1: Initialized pretrained MoE LLM Information.
> >
> > We confirm that the initialized pretrained MoE LLM used for the main experiments is **DeepSeek-V2 Lite**. We have explicitly added this specification to the **Section 4.2** in the revised paper to ensure reproducibility.
> >
> > ----
> >
> > > Question3: More evaluation results with extended competitors and benchmarks.
> >
> > We thank the reviewer for the suggestion to conduct a more comprehensive evaluation, we have expanded our Spoken Question Answering (SQA) evaluation to include **Speech-to-Text (S-T)** results for LlamaQ and added **WebQ**[6] (S-T and S-S) as a new benchmark. We also added comparisons against recent strong baselines: **MinMo**[7] and **Llama-Omni2**[8].
> >
> > **Table 3: Extended Spoken Question Answering (SQA) Performance**
> >
> > | Model | LlamaQ (S-T) | LlamaQ (S-S) | TrivialQA (S-T) | TrivialQA (S-S) | WebQ (S-T) | WebQ (S-S) |
> > | :--- | :---: | :---: | :---: | :---: | :---: | :---: |
> > | Audio-LM | - | 7.00 | - | - | - | 2.3 |
> > | SpeechGPT | 21.6 | 5.4 | 14.8 | 8.2 | 6.3 | 5.5 |
> > | SpiritLM | 40.1 | 38.7 | 6.5 | 4.2 | 12.3 | 9.6 |
> > | Moshi | 60.2 | 57.3 | 22.8 | 7.3 | 26.6 | 9.2 |
> > | Phi-4-Multimodal | 68.3 | / | 42.1 | / | 47.5 | / |
> > | Qwen2-Audio | 55.4 | / | 38.5 | / | 41.2 | / |
> > | MinMo | **76.5** | **63.8** | 40.1 | 25.5 | 55.0 | 39.9 |
> > | Llama-Omni2 | 70.3 | 60.7 | 35.3 | 23.9 | 40.4 | 37.1 |
> > | **MoST (Ours)** | 74.8 | 62.6 | **43.5** | **32.1** | **58.2** | **44.7** |
> >
> > MoST achieves state-of-the-art performance on **TrivialQA (S-S & S-T)** and **WebQ (S-S & S-T)**, and remains highly competitive on LlamaQ. Notably, MoST outperforms MinMo on WebQ (S-S) by **+4.8%** and on WebQ (S-T) by **+3.2%**. This demonstrates that our MAMoE architecture generalizes exceptionally well to diverse spoken query tasks compared to up-to-date strong speech-text LLMs.
> >
> > > Question4: Fix of minor issues.
> >
> > We thank the reviewer for the detailed eye. We have:
> > - fixed the citation format (using `\citep`) in **the Section 6.1**,
> > - clarified the MoST-Style Upcycling setup in **the Section 6.1**,
> > - corrected the legend in **Figure 5** to clearly label the ablation as "MoST-NoShared"
> >
> > in the revised manuscript.
> >
> > **References:**
> >
> > [1] DeepSeek-AI, et al. "DeepSeek-V2: A Strong, Economical, and Efficient Mixture-of-Experts Language Model." arXiv preprint arXiv:2405.04434 (2024).
> >
> > [2] Dan Hendrycks, et al. "Measuring Massive Multitask Language Understanding." arXiv preprint arXiv:2009.03300 (2021).
> >
> > [3] Mandar Joshi, et al. "TriviaQA: A Large Scale Distantly Supervised Challenge Dataset for Reading Comprehension." arXiv preprint arXiv:1705.03551 (2017).
> >
> > [4] Karl Cobbe, et al. "Training Verifiers to Solve Math Word Problems." arXiv preprint arXiv:2110.14168 (2021).
> >
> > [5] Mark Chen, et al. "Evaluating Large Language Models Trained on Code." arXiv preprint arXiv:2107.03374 (2021).
> >
> > [6] Jonathan Berant, et al. "Semantic Parsing on Freebase from Question-Answer Pairs." Proceedings of the 2013 Conference on Empirical Methods in Natural Language Processing (2013).
> >
> > [7] Qian Chen, et al. "MinMo: A Multimodal Large Language Model for Seamless Voice Interaction." arXiv preprint arXiv:2501.06282 (2025).
> >
> > [8] Qingkai Fang, et al. "LLaMA-Omni2: LLM-based Real-time Spoken Chatbot with Autoregressive Streaming Speech Synthesis." arXiv preprint arXiv:2505.02625 (2025).

---

> > > ### Author Response · Authors · 2025-11-28
> > > **Summary of Rebuttal and New Experiments**
> > >
> > > Dear Reviewer pG6E,
> > >
> > > As the rebuttal period concludes, we would like to sincerely thank you again for your thoughtful review and constructive feedback, which has significantly strengthened our paper. We briefly summarize how we addressed each of your concerns:
> > >
> > > 1. **Expert Initialization (W1)**: We clarified that in standard MoE models, experts are randomly initialized without semantic clustering, making index-based partition statistically neutral. Stage 1 training specializes these experts, while shared experts preserve general capabilities. We acknowledge this limitation and have added **discussion of more sophisticated initialization strategies** in **Conclusion and Appendix D**.
> > >
> > > 2. **Text-only Evaluations (W2 & Q2)**: We provided comprehensive results on **MMLU, TriviaQA, GSM8K, and HumanEval** (Table 1 in rebuttal). MoST shows only modest degradation on MMLU (-2.9%) while achieving significant improvements on reasoning (+25.8% GSM8K) and coding (+28.3% HumanEval), substantially outperforming other speech-native models.
> > >
> > > 3. **Data Efficiency Claims (W3)**: We clarified that "efficiency" encompasses both **data source efficiency** (exclusively open-source data) and **synthesis efficiency** (self-generated instruction data). We provided detailed **training recipes** with exact data mixing ratios (Table 2 in rebuttal).
> > >
> > > 4. **Base Model Specification (Q1)**: Confirmed **DeepSeek-V2 Lite** as our base mode
> > >
> > > 5. **Extended Evaluations (Q3)**: Added **LlamaQ (S→T)** results, **WebQ benchmark**, and comparisons with MinMo and Llama-Omni2 (Table 3 in rebuttal), demonstrating MoST's state-of-the-art performance.
> > >
> > > 6. **Minor Issues (Q4)**: All corrections have been made in the revised manuscript.
> > >
> > > We hope our responses and additional experiments have adequately addressed your concerns. Thank you again for your valuable time and feedback.

---

### Author Response · Authors · 2025-11-26
**Overall Response to All Reviewers**

Dear Reviewers pG6E, Mkjs, and cs1j,

We sincerely thank all reviewers for the thorough, constructive, and insightful feedback. Your thoughtful comments have significantly strengthened our paper, and we deeply appreciate the time and expertise you invested in reviewing our work. We are particularly encouraged by the recognition of **MAMoE's novelty** (Reviewers pG6E, cs1j), **the rigor of our experimental validation** (Reviewer pG6E), and **our commitment to open-source research** (Reviewers Mkjs, cs1j).

---

## **Summary of Key Revisions**
Based on your valuable feedback, we have made substantial improvements to the paper across three main dimensions:

### **1. Enhanced Experimental Evaluation**

**Extended Benchmark Coverage:**

- Added comprehensive **text-only evaluations** (MMLU, TriviaQA, GSM8K, HumanEval) to assess impact on foundation model capabilities (**Appendix E**)
- Expanded Spoken Question Answering evaluation to include **WebQ benchmark** and **Speech-to-Text (S→T) results for LlamaQ**  (**Section 5.4**)
- Added comparisons with recent concurrent works: **MinMo** and **LLaMA-Omni2** (**Section 5**)

**Results Summary**: MoST achieves state-of-the-art or highly competitive performance across all benchmarks while maintaining 93% of original MMLU score and demonstrating significant improvements in math (GSM8K: +25.8% over base model) and coding tasks (HumanEval: +28.3% over base model).


### **2. Enhanced Methodological Clarity**

**Architectural Justification:**

- Clarified that MAMoE outperforms tradtional dense/MoE models in processing speech-text mixed data with ablation study results in **Section 6.1,6.2**
- Clarified that **routing is dynamic within modality-specific groups**, not static tag-based assignment
- Provided empirical evidence showing MAMoE achieves **lower routing entropy and better load balancing** compared to vanilla MoE

**Training Protocol Transparency:**

- Explicitly specified **base model initialization** (DeepSeek-V2 Lite) in **Section 4.2**
- Added comprehensive **data usage and training recipe** with exact data mixing ratios, batch sizes, and step counts (**Table 3 in Appendix A**)
- Detailed **data source efficiency** advantages over proprietary-data baselines

**Initialization Strategy Discussion:**

Acknowledged that index-based expert partition, while effective, may not be optimal
Added discussion of future directions including clustering-based and knowledge-preserving partitioning strategies (**Conclusion and Appendix D**)

### **3. Technical Corrections and Improvements**

- Fixed citation formatting throughout the paper (**Section 6.1**)
- Corrected **Figure 5** legend to accurately label ablation variants
- Enhanced clarity of MoST-Style Upcycling experimental setup (**Section 6.1**)
- Added explicit analysis of **MoE's impact on text foundation model capabilities with controlled experiments**

---

## **Looking Forward**
We believe the revisions have significantly strengthened the paper's contribution to the speech-text LLM community. The additional experiments and clarifications directly address all major concerns while maintaining the paper's core novelty and impact. We remain committed to the full open-source release of MoST to facilitate future research in efficient and effective multimodal models.

We welcome any further questions or suggestions for improvement and remain available for continued discussion during the review period.

Thank you once again for your invaluable feedback and guidance.

The Authors

---

### Author Response · Authors · 2025-12-02
**Official Comment to Area Chair: Summary of Rebuttals, New Experiments, and Key Contributions (1/3)**

Dear Area Chair,

We fully understand the challenging circumstances resulting from the recent data leak and the significant additional workload this places on the program committee. We sincerely appreciate you stepping in to oversee the decision process for our submission. We remain committed to maintaining the scientific integrity of ICLR and hope this summary assists you in your evaluation.

During the rebuttal period, we engaged extensively with Reviewers **pG6E**, **Mkjs**, and **cs1j**. We conducted significant new experiments—including adding new baselines (MinMo, Llama-Omni2), performing text-only foundation model benchmarks, and running controlled architectural ablations—to address their concerns.

Below, we summarize our exchange with each reviewer, followed by a reiteration of our key contributions.

---

> ### Author Response · Authors · 2025-12-02
> **Official Comment to Area Chair: Summary of Rebuttals, New Experiments, and Key Contributions (2/3)**
>
> ## **Summary of Reviews and Rebuttals**
>
> ### **Reviewer pG6E**
> **Key Focus**: Expert initialization, text-only performance, and data efficiency details.
> | Concern | Our Response | Key Evidence & Findings |
> |---------|--------------|------------------------|
> | **W1: Index-based 50% expert partition may lose text knowledge** | Standard MoE experts are randomly initialized without semantic clustering, making index-based partition statistically neutral. **Stage 1 training specializes experts while shared experts preserve general capabilities.** We acknowledge this limitation and discuss future directions. | Discussion added in **Conclusion & Appendix D**; Text benchmarks show only 2.9% MMLU degradation while improving GSM8K (+25.8%) and HumanEval (+28.3%) |
> | **W2: Missing text-only evaluations** | Conducted comprehensive text benchmarks (MMLU, TriviaQA, GSM8K, HumanEval). MoST preserves 93% of original text performance and significantly outperforms other speech-native models. | **Table 1 (Rebuttal)**: MoST achieves MMLU 55.4, GSM8K 66.9, HumanEval 58.2 vs. SpiritLM (36.9, 21.5, 9.7) and Moshi (49.8, 40.3, 25.4) |
> | **W3: Efficiency claims lack quantifiable metrics** | Clarified two dimensions: (1) *Data source efficiency*—exclusively open-source data; (2) *Synthesis efficiency*—self-generated instruction data without expensive annotation. | **Table 2 (Rebuttal)**: Detailed training recipe with exact data mixing ratios (ASR: 0.4, TTS: 0.4, LM: 0.2 for Stage 1) |
> | **Q1: Base model specification** | Confirmed DeepSeek-V2 Lite as the initialized pretrained MoE LLM. | Explicitly added to **Section 4.2** in revision |
> | **Q3: LlamaQ (S→T) and extended benchmarks** | Added LlamaQ (S→T), WebQ benchmark, and comparisons with MinMo and LLaMA-Omni2. | **Table 3 (Rebuttal)**: MoST achieves SOTA on TriviaQA (S-S: 32.1, S-T: 43.5) and WebQ (S-S: 44.7, S-T: 58.2), outperforming MinMo on WebQ by +4.8% (S-S) |
> | **Q4: Minor issues (citations, figure labels)** | Fixed all citation formatting, clarified MoST-Style Upcycling setup, corrected Figure 5 legend. | Corrections in revised **Section 6.1** and **Figure 5** |
>
> ---
>
> ### **Reviewer Mkjs**
> **Key Focus**: Motivation for MoE, added concurrent baselines, and open-source clarification.
> | Concern | Our Response | Key Evidence & Findings |
> |---------|--------------|------------------------|
> | **W1: Motivation for modality-aware router unclear** | Dense models and standard MoEs suffer from modality interference when processing distinct representations. **Our ablation study directly compares MAMoE, Traditional MoE, and Dense baselines with identical initialization.** | **Section 6.1, Figure 4**: MAMoE achieves lower loss and better performance across all 4 tasks vs. Traditional MoE and Dense baselines; Literature support from MoMa (Lin et al., 2024) |
> | **W2: Missing evaluations (LlamaQ S2T, WebQ, text-based, base model info)** | Extended evaluation to include all requested benchmarks and clarified base model (DeepSeek-V2 Lite). | **Tables 2-5 (Rebuttal)**: Comprehensive results across ASR, TTS, Audio LM, SQA, and text benchmarks |
> | **W3: Missing comparisons with MinMo, LLaMA-Omni2; open-source concerns** | Added comparisons with both concurrent works. MoST provides **the most complete open-source contribution** in this domain. | **Table 6 (Rebuttal)**: MoST is the *only* model releasing Train Code + Eval Code + Inference Code + Dataset & Pipeline; MoST outperforms LLaMA-Omni2 on all SQA metrics |
> | **Q1: MoE's impact on text foundation model** | Controlled experiment comparing Dense, Traditional MoE, and MoST-Style Upcycling from identical Llama 3.2 3B initialization. | **Table 7 (Rebuttal)**: Dense suffers severe forgetting (MMLU -17.1 pts); MoST-Style preserves 93-94% of text capability while achieving best speech performance |

---

> ### Author Response · Authors · 2025-12-02
> **Official Comment to Area Chair: Summary of Rebuttals, New Experiments, and Key Contributions (3/3)**
>
> ### **Reviewer cs1j**
> **Key Focus**: Routing mechanism design (Static vs Dynamic) and performance margins.
> | Concern | Our Response | Key Evidence & Findings |
> |---------|--------------|------------------------|
> | **W1 & Q1: Concern that tag-based routing resembles a "Two-Tower" architecture and underutilizes MoE dynamism** | This concern is a **misunderstanding**. Our routing is **dynamic *within* modality-specific groups**—only group assignment is deterministic; expert selection uses content-based softmax scores. The model learns specialized experts within groups. | **Algorithm 1 (Steps 4-5), Section 3.3.3**: Dynamic intra-group routing; **Figure 6 (Right)**: Routing entropy decreases during training, proving content-based specialization |
> | **W2: Limited performance margins** |We highlighted that while single-metric gains may vary, MoST is the **only model** to achieve Top-3 performance across **all** benchmarks (ASR, TTS, Audio LM, SQA). All results achieved with 100% open-source data vs. proprietary data baselines. | Cross-dataset ASR: VoxPopuli 6.2% vs 7.1% (Qwen2-Audio); Spoken QA (TriviaQA S→S): 32.1% vs ~7% (Moshi)—representing 4-8× improvement |
> | **Q2: Missing MoE baseline comparisons** | MoST is the first MoE-based speech-text LLM; we ran a new controlled experiment comparing **MoST-Style Upcycling** vs. **Traditional Sparse Upcycling**| **Section 6.2, Figure 5**: MAMoE outperforms Vanilla MoE on all tasks; **New controlled experiment**: MoST-Style outperforms Traditional MoE by +5.8 avg pts on text and 7-22% on speech |
>
>
> ---
>
> ## **Key Contributions of Our Work**
> Based on the revised manuscript and new experimental data, **MoST** offers the following distinct contributions to the multimodal community:
>
> ### 1.  **Modality-Aware Mixture of Experts (MAMOE) Architecture**
>
>
> We introduce a **novel MoE architecture** combining three components: modality-specific expert groups, cross-modal shared experts, and a modality-aware router.
>
> - **Dynamic routing within groups**: While group assignment is deterministic (text tokens → text experts, audio tokens → audio experts), expert selection *within* each group uses content-based softmax scores (Algorithm 1, Steps 4-5)
> - **Empirically validated**: MAMoE achieves lower routing entropy and better load balancing compared to vanilla MoE (Figure 6), with consistent outperformance across ASR, TTS, Audio LM, and SQA tasks (Figure 5)
>
>
> ### 2. **Efficient LLM-to-Speech-Text Transformation Pipeline**
>
> We develop a **data-efficient two-stage training protocol**:
> - *Stage 1*: Cross-Modal Post-Training on ASR/TTS data (500k steps)
> - *Stage 2*: Mixed Instruction Fine-Tuning with speech-text instructions (10k steps)
>
> Key features include:
> - **Exclusive use of open-source data**: LibriHeavy, Common Voice, VoxPopuli, SmolTalk—no proprietary datasets
> - **Self-synthesis approach**: Uses Stage-1 trained MoST to generate speech instruction data, avoiding expensive manual annotation
>
>
> ### 3. **State-of-the-Art or Competitive Performance Across All Benchmarks**
>
> | Task Domain | Key Results |
> |-------------|-------------|
> | **ASR/TTS** | SOTA TTS on LS-Clean (6.0%), VoxPopuli (10.1%), Common Voice (11.5%); competitive ASR |
> | **Audio Language Modeling** | Best on sWUGGY (75.28%) and sTopic-StoryCloze (83.64%); avg 71.94% |
> | **Spoken QA** | SOTA on TriviaQA S→S (32.1%), WebQ S→T (58.2%) and S→S (44.7%) |
> | **Text Benchmarks** | 93% MMLU retention; +25.8% GSM8K, +28.3% HumanEval vs. base model |
>
>
> ### 4.  **Full Open-Source Commitment**
>
> To our knowledge, MoST is the first fully open-source speech-text LLM built on a Mixture of Experts architecture. We release the **model weights, training code, inference code, and the curated training datasets**, filling a critical gap in reproducibility for the field.
>
> ---
> ## **Conclusion**
> We believe our extensive rebuttals, including new experiments with controlled comparisons, extended benchmarks (MinMo, LLaMA-Omni2, WebQ, LlamaQ S→T), comprehensive text evaluations, and detailed clarifications on architectural design, have thoroughly addressed all reviewer concerns.
>
> Thank you again for your time and service to the community.
>
> Sincerely,
>
> **Authors of Paper 11337**

---

### Meta-Review · Area_Chair_sR7V · 2025-12-15

**Summary:**

This paper introduces MoST, a speech-text large language model built on a Modality-Aware Mixture of Experts (MAMoE) architecture. The core idea is to extend a pretrained MoE language model with modality-specific experts for speech and text, plus shared experts for cross-modal interaction. The authors employ a two-stage post-training pipeline (ASR/TTS adaptation and mixed instruction fine-tuning) and evaluate MoST on ASR, TTS, audio language modeling, and spoken QA tasks. The paper addresses an important challenge in unifying speech and text within large language models and proposes a structured solution via modality-aware MoE. The design is appealing and experiments are reasonably thorough, but the contribution is somewhat incremental and lacks strong theoretical justification. Missing evaluations, unclear efficiency analysis, and limited novelty in routing design reduce the strength of the claims.

**Reviewer Concerns:**

I think some of reviewer concerns have been addressed.

**Reviewer Scores:**

I think it is possible that the reviewer would have changed their score.

---

### Decision · Program_Chairs · 2026-01-26

Reject